# Food Safety Aspects of Breeding Maize to Multi-Resistance against the Major *(Fusarium graminearum*, *F. verticillioides*, *Aspergillus flavus*) and Minor Toxigenic Fungi (*Fusarium* spp.) as Well as to Toxin Accumulation, Trends, and Solutions—A Review

**DOI:** 10.3390/jof10010040

**Published:** 2024-01-04

**Authors:** Akos Mesterhazy

**Affiliations:** Cereal Research Non-Profit Ltd., Alsokikotosor 9, 6726 Szeged, Hungary; akos.mesterhazy@gabonakutato.hu

**Keywords:** *Fusarium graminearum*, *F. verticillioides*, *Aspergillus flavus*, ear rot resistance, resistance to toxin accumulation, multi-toxin, breeding for complex resistance, food safety, climate control and resistance, integrated disease and toxin control

## Abstract

Maize is the crop which is most commonly exposed to toxigenic fungi that produce many toxins that are harmful to humans and animals alike. Preharvest grain yield loss, preharvest toxin contamination (at harvest), and storage loss are estimated to be between 220 and 265 million metric tons. In the past ten years, the preharvest mycotoxin damage was stable or increased mainly in aflatoxin and fumonisins. The presence of multiple toxins is characteristic. The few breeding programs concentrate on one of the three main toxigenic fungi. About 90% of the experiments except AFB1 rarely test toxin contamination. As disease resistance and resistance to toxin contamination often differ in regard to *F. graminearum*, *F. verticillioides*, and *A. flavus* and their toxins, it is not possible to make a food safety evaluation according to symptom severity alone. The inheritance of the resistance is polygenic, often mixed with epistatic and additive effects, but only a minor part of their phenotypic variation can be explained. All tests are made by a single inoculum (pure isolate or mixture). Genotype ranking differs between isolates and according to aggressiveness level; therefore, the reliability of such resistance data is often problematic. Silk channel inoculation often causes lower ear rot severity than we find in kernel resistance tests. These explain the slow progress and raise skepticism towards resistance breeding. On the other hand, during genetic research, several effective putative resistance genes were identified, and some overlapped with known QTLs. QTLs were identified as securing specific or general resistance to different toxicogenic species. Hybrids were identified with good disease and toxin resistance to the three toxigenic species. Resistance and toxin differences were often tenfold or higher, allowing for the introduction of the resistance and resistance to toxin accumulation tests in the variety testing and the evaluation of the food safety risks of the hybrids within 2–3 years. Beyond this, resistance breeding programs and genetic investigations (QTL-analyses, GWAM tests, etc.) can be improved. All other research may use it with success, where artificial inoculation is necessary. The multi-toxin data reveal more toxins than we can treat now. Their control is not solved. As limits for nonregulated toxins can be introduced, or the existing regulations can be made to be stricter, the research should start. We should mention that a higher resistance to *F. verticillioides* and *A. flavus* can be very useful to balance the detrimental effect of hotter and dryer seasons on aflatoxin and fumonisin contamination. This is a new aspect to secure food and feed safety under otherwise damaging climatic conditions. The more resistant hybrids are to the three main agents, the more likely we are to reduce the toxin losses mentioned by about 50% or higher.

## 1. Introduction

Ear rots due to toxigenic fungi cause significant damage in maize. They cause losses in maize grain yield, and most of the damage is caused by the presence of mycotoxins. Some of these toxins have gained extra attention in recent years, as the warming of the climate—namely, warmer, and hotter vegetative seasons—is expected to increase the risk of contamination by fumonisins and (mostly) aflatoxins. Maize is one of the most-exposed crops to mycotoxins; except for patulin, all other major mycotoxins, e.g., aflatoxins (AFB1), ochratoxin A (OTA), deoxynivalenol (DON), zearalenone (ZEN), and fumonisins (FUMs), occur on maize at significant levels in the European and global contexts [1], with certain regional differences. Therefore, in this crop, it is a significant challenge to reduce not only the presence of individual toxins but also (if possible) to keep all toxins under the official limits for food and feedstuffs, resulting in a very complex task. Despite this situation, the literature on Fusarium in wheat is about three-fold greater than that in maize. An increasing amount of information has indicated that multi-toxin contamination is a serious food safety challenge; as such, research should be urgently conducted to provide solutions to keep such contamination under control [2].

Considering the information we have, it becomes clear that the generally accepted view of common resistance to the different pathogens may not be the actual case. Earlier, it was supposed, based on natural ear rot, that the breeding for resistance can be successful; some generally valid resistance may arise against the different pathogens in maize. Later, the problem was differentiated to Fusarium ear rot (FER) and Gibberella ear rot (GER), followed by a number of different *Fusarium* species [3]. However, no data indicate that resistance to one *Fusarium* pathogen automatically means resistance to another *Fusarium* sp. or all. However, we have examples of several hybrids having common resistance to *F. graminearum* and *F. culmorum*, *F. graminearum* and *F. verticillioides*, and *F. verticillioides* and *A. flavus*. More than two toxic species were not tested. The aflatoxin producing *Aspergillus* spp. has the same problem; resistance to *A. flavus*, *A. parasiticus*, or other *Aspergillus* spp. does not necessarily result from the same genetic background. The same is true for the possible resistance relationships between *Fusarium* and *Aspergillus* species. Another problem is that, based only on several tests, resistance to disease and resistance to toxin accumulation were presumed to be automatically coupled, meaning that a higher resistance automatically relates to lower toxin contamination, and, so, the two traits may be considered to be equivalent [4,5].

However, the actual case seems to differ; in this review, this is a highly important point of view. Without resistance data (as measured according to symptom severity or other means), we cannot say anything regarding the resistance to toxin accumulation, as we do not have an objective point of comparison; therefore, the need for a complex approach is inevitable, as has been stressed by Campbell and White [6]. This point is therefore important, as we do not speak here about the resistance of plants to toxins, or the resistance and susceptibility of humans and animals to toxin exposure; we speak in this context only about the resistance to toxin accumulation in diseased plants. The other two topics are also significant; however, they are subjects that need to be explored in future research. This agrees well with the idea of Bhatnagar et al. [7]. The third problem arises from the need for food and feed safety. From multi-toxin tests, it becomes clear that many toxins may exist in a given sample. As the same toxin can be produced by many different species, it is often not clear which fungus contributes a given toxin. This is also true in the other direction: when identifying a fungus, 5–10 different toxins may also be identified, considering masked, acetylated, and other variants. This is a different perspective, making identification a much more complicated task than previously thought. The last ten years have brought many new ideas and developments that can be summarized in the hope of finding new interpretations and solutions for the key problems that we are currently facing.

This problem is very relevant, as we are presently facing many new challenges. We summarized the earlier literature in 2012 [8], in which we stated that most of the inbreds and hybrids are susceptible or very susceptible to ear rots. This is true also today. In the variety brochures, the resistance to toxigenic fungi or toxins is rarely mentioned as a problem. Despite growing research, a significant decrease in mycotoxin contamination has not been observed in the last decade, and, considering aflatoxins and fumonisins, an increase has been forecast due to increasing drought and heat-shock [9], as the case in Hungary shows [10]. Considerable progress has been made in the multi-toxin field in terms of breeding relations and alternative solutions, such as the use of the Bt gene complex from *Bacillus thuringiensis* or atoxic *A. flavus* strains to replace aflatoxin-producing strains [11]. A key question is whether a reduction in aflatoxin contamination is enough to secure the necessary food safety for both consumers and their animals or not. Knowing the multi-toxin results in corn, the answer is surely not. Kaaler et al. [12], in their latest review on maize about the control of the aflatoxins, mention the breeding for resistance as a possible preventive way.

Methodical problems should also be considered. Table 1 lists various inoculation methods and their evaluation in the published literature. In the first case, only one inoculum was used in every test, either a pure isolate or a mixture of isolates. In the second case the conidium concentration was regulated, but no aggressiveness test was conducted, and, in the best case, prior experience was referred to. Different scales have been used for the evaluation of ear rot. The maximum and minimum values were given in some papers but are often absent. We speak about food and feed safety, but most papers discussed only visual symptoms; for example, the fungal mass determined based on quantitative PCR data or other means were reported in several cases, and exceptionally, like in wheat, toxin data showed rather close correlations between fungal mass and toxin content [13]. Toxin data are included rather exceptionally. In this respect, no significant change was observed over the past 12 years.

As our previous review was written in 2010–2011 [8], we concentrated on the last 13–14 years in order to observe the more recent tendencies and solutions. In this way, we hoped to determine the reasons underlying the slow progress in the field and find a way to improve the efficacy of research and breeding work. It is necessary to understand the possibilities for better agronomy, plant protection, and irrigation technology in order to better support the corn plants, while producing the lowest possible amount of toxins. Climate change on maize is often discussed, but the roles of more-resistant plants in terms of balancing the increase in toxin contamination are not frequently mentioned, even though this line of research seems to have a prosperous outlook. Therefore, more careful analyses are required.

A significant part of the published literature used inbred lines for experimental purposes, which is good for scientific research. Meanwhile, hybrid testing is generally considered to be less interesting; although, maize producers would be interested, but only without the food safety risk when growing hybrids. Of course, there are several publications that exist [47,49]. For this reason, the central problem seems to be the official control of the resistance of hybrid candidates during the varietal registration process, as this seems to be the only way to control food safety much better than what is performed at present. This is valid also for the commercially grown hybrids, too. As breeders concentrate on yielding ability, the resistance background of the hybrids remains mostly in shadow. A vast majority of the tested international hybrids are susceptible or highly susceptible to one or more major toxigenic fungi [10,50,51,52]. This is proof that this is not the central problem of the maize breeding. As scientific background is also inadequate, the knowledge is often insufficient, and the investment into the resistance seems to be problematic, as it might not pay off. So, other practices come to the foreground in the hope that the toxin contamination can be reduced. Many articles concentrate on the advantage of practices of reducing toxin concentration coming from the contaminated field or detected in storage bins. This is called toxin management and involves what to do with the lots with a high concentration of one or more mycotoxins.

We conducted fungicide tests against ear rot fungi combined with artificial inoculation; however, in the highly susceptible hybrids, only a moderate (and often nonsignificant) reduction in toxin contamination was observed, even with the best fungicides used in the wheat Fusarium head blight (FHB) control [53,54]. Even when we observed up to a 50–70% reduction in wheat grain yield, the remaining toxin contamination remained three-to-five-fold higher than the allowable limit. In the author’s opinion, effective control should mean that the mycotoxin contamination is suppressed below a binding or suggested limit. As the higher resistance and fungicide treatment were effective, this should serve as an example for relevant research in maize. *Therefore*, *highly susceptible hybrids should be withdrawn from commercial production, or*, *at least*, *their registration should be inhibited.*

The main objective of this paper is to summarize the new results in the testing methodology, resistance research, variety testing, and genetic aspects to present updated knowledge to make significant improvements in food and feed safety in maize production. This seems to be a good outlook, with multiple perspectives.

## 2. Ear Rot, Mycotoxins, and Losses in Maize Production

What is the significance of toxigenic fungi? The FAO has estimated that mycotoxin contamination is above the relevant limit in 25% of harvested grains [41]. Eskola et al. [55] collected globally available data and confirmed that this figure of 25% mycotoxin reduction seems appropriate. However, they could not differentiate between preharvest and postharvest origin; therefore, relevant solutions could not be suggested. The global maize harvest is 1060 million metric tons [56]. Considering this rate, *the yearly global mean mycotoxin loss for maize can be estimated with the 25% FAO rate, which is about 265 million metric tons (MTT)*. Mesterhazy et al. [57] assessed global mycotoxin contamination for harvested grains at 10% (210 million metric tons), with preharvest losses (less grain yield) at 2–4% and the storage losses around 20% (440 million metric tons). A great part of these losses was caused by mycotoxins; however, the actual rate is not known. The 33% loss due to mycotoxins during storage seems to be a conservative estimation. However, it is hard to say from [41,55] which part of the loss is at preharvest, post-harvest, or both. When mycotoxin control at harvest is globally introduced and control toxin data at the end of storage are provided, we will have more exact data. When we incorporate these three sources, it is expected that the total damage of about 20% (220 MMT) will not differ greatly from the estimates of the FAO and Eskola et al. [55], but the origin is clearer, and the tasks can be better identified to decrease losses. The data for maize, based on global statistics [57], are provided in Table 2.

The preharvest loss can be estimated as one-third of the yield loss and, so, the 1060 million metric tons harvested should be increased to 1590 million metric tons. The mycotoxin contamination causes quality loss, and the otherwise existing grain suffers quality loss. The same is the case with the storage loss: this mostly degraded grain is not suitable for food or feed. In Hungary, the national mean grain yield was in 2021 6.18 t/ha, and the production capacity (yield + preharvest loss) is 9.27 t/ha; however, the testing of varieties indicated an average yield of 15.03 t/ha in the best-yielding experiments for 18 early (FAO 300–399) cultivars, with a maximum of 16.37 t/ha and a minimum of 13.5 t/ha (GOSZ-VSZT post-registration tests, 2021) [58]. For the FAO 400–499 group, the best location produced 16.07 t/h as the mean, the highest yielding hybrid gave a 16.85 t/ha grain yield, and 14.86 t/ha was the minimum [58]. The worst location with the same hybrids yielded only 4.96 t/ha. This means that, above the production capacity in Hungary, we have at least 50% reserve [58]. In 2020, the national grain yield was 8.6 t/ha, and the grain yield capacity was 12.9 t/ha. The best location gave for the FAO 300–399 group 15.06 t/ha, with 16.02 as the maximum and 13.58 t/h as the minimum, so the mean was higher, i.e., 16% better than the grain yield capacity. The best hybrid surpassed the grain yield capacity by 24%. In the FAO group 400–499, the best location yielded 15.66 t/ha, 21% greater than the grain yield capacity; in the FAO group, the mean was 15.35 t/ha (+19%), the best yielded 16.32 (+27%), and the worst yielded 14.58, with 13% greater than the national yielding capacity [59]. I mention that only because, in 2022, the national mean was only 3.4 t/ha because of the draught, and the aflatoxin data were in the high for natural contamination, i.e., between 96 and 1126 mg/kg (Mesterhazy unpublished). This was the first year since 2014 when all hybrids contained aflatoxin in levels above the limit. In spite of this, we have a 11.77-fold difference between the most and least resistant hybrids. Even under these conditions, it has a sense to prefer the lower toxin concentration provided by hybrids. This is a serious warning. We have to pay attention. We have data for at least the last 10 years, and every time, with differences, the main tendency is this. This would be important to see in each maize production country, and we could encourage international cooperation to make a global analysis. The conclusion is that increasing of the genetic grain yield alone does not solve any problem, as the losses remain at about 62% compared to the global grain yield capacity [57]. *This maize production system is not sustainable and results in unacceptable losses and low efficacy during the production process. Instead*, *significant improvement in the adaptation ability (biotic and abiotic) of hybrids and better agronomy supported by biological tillage can help to increase grain yields. Furthermore*, *renewal of the harvest–storage can contribute to increasing the healthy grain mass used.*

Knowledge of the origin of the various mycotoxins is important, as the solution will differ according to the source. For the preharvest losses, resistance and agronomy should provide solutions; meanwhile, regarding the storage problem, the first important condition is that only healthy grain should be used that for long storage, and the storage facility must also provide adequate conditions to preserve the quality of the stored grain. Compared to preharvest problems, storage problems can be resolved much easier; however, without excellent preharvest control, the prevention of toxin contamination in the storage facilities is a nearly hopeless task. In an eight-year period, we found three aflatoxin years, with 2022 being the fourth such year [51]. As such, it is clear that, in Hungary, preharvest-origin aflatoxin contamination is present by up to more than 2000 mg/kg and strongly differs between hybrids. We recorded fumonisins in two investigated years (2014 and 2021), while in 2022, there was also a large presence of fumonisin. A significant amount of DON was recorded in three years, while, in 2022, no significant preharvest DON contamination was found, and only two years were without significant DON, fumonisin, and aflatoxin production across hybrids [51]. These data refer to the means in hybrids and, so, in years with lower mean levels, several hybrids may have still surpassed the relevant limit values. It has been much less considered that maize presents significant mycotoxin contamination problems [1]; therefore, it is not surprising that the losses in maize are generally much higher in terms of general grain mycotoxin contamination when compared to other cereals. The differences between hybrids in resistance to disease and resistance to toxin accumulation are highly significant; therefore, the investment in breeding focused on resistance, and the reduction of exposure to toxins is highly profitable and should be considered more seriously than it is at present. Considering the existing knowledge, a higher resistance does not necessarily mean a lower grain yield: among the top-yielding hybrids (the grain yield tests were conducted in the Variety Testing Office (Nebih) for registration. The postregistration tests were performed by the cereal producers, seed companies, and the National Agrar Chamber for the registered hybrids also performed by Nebih; both highly resistant and highly susceptible varieties can be found [10,60]. However, high toxin contamination can cause a complete financial loss.

AFB1, fumonisin, and DON contents strongly respond to increasing temperatures and drought conditions [1,61,62], and the combination of warm weather and higher precipitation favors DON-producing fungi, as well as many other toxigenic fungi, leading to high toxin accumulation rates [10,51]. Toxin forecasts are based on meteorological factors [9], with a 2.5 °C increase in temperature for the vegetative period throughout the whole Carpathian Basin, Lower Austria, and South Germany, as well as a 5 °C increase for Helsinki, potentially threatening higher aflatoxin contamination rates. At present, there exist no reliable resistance data for ear diseases that could be used in climate models. *This is therefore important*, *as nobody can work with non-existing data*; *even the general train of thought agrees with the idea that*, *in theory*, *resistance is the most promising solution.* If the present resistance levels of hybrids do not change, we will find that such forecasts were correct in the next 20 years. Thus, it is our common duty to test the possibilities of breeding and, if they can be proven, to determine how we can utilize them to control the negative effects due to increasing temperatures. This, however, is only one part of the problem. We speak about possibilities for the 10–15-year horizon, and the forecasts seem to otherwise be fair. Earlier in Hungary, there were not any problems with aflatoxin, unlike the situation at present [51,59]. Very large maize-producing areas are now under toxin pressure now, like the Mediterranean, Africa, Southeast USA, and South Asia, which are presently facing significant problems, and thus need help as soon as possible.

It is well known that insect injuries may significantly increase fumonisin and aflatoxin contamination [3,63]. This scenario is not treated in this paper; however, we do mention the resistance relations of this problem regarding breeding when no *Bt* gene GM solution is possible. We know that susceptible hybrids can be severely infected without insect damage. In this case, the *Bt* gene does not help too much. Therefore, a lower toxin contamination could be postulated when higher resistance to ear rot would be the case.

In this review, we discuss and evaluate problems related to the disease-causing organisms, the preharvest and postharvest mycotoxin situation, multi-toxin problems together with the mycotoxin contents in blood and urine, the resistance of maize to different toxigenic species and the nature of this resistance to these fungi, and methodical problems. Breeding aspects are also analyzed. In the last ten years, an increasing number of publications have dealt with the relationships between climate change and epidemics that increase exposure to toxins. In 2010–2011, the aflatoxin problem was far less severe than that at present [8], and, so, the aflatoxin problem will be considered with higher weight, due its significance. In particular, the aflatoxin epidemic in 2012 alarmed experts and growers. It seems that plant breeders must thoroughly think about their contributions and responsibilities to the reduction of mycotoxin pressure.

## 3. Ear Rot-Causing Agents and Toxin Relations, General View

Many *Fusarium* species are known to infect maize ears [8], with most species having been reported from Europe (see Table 1, 2012), Canada, Mexico, and the USA, reporting 1–4 *Fusarium* species. In the USA and Canada, *F. graminearum* occurs more often in the northern parts, while on the connecting southern areas, *F. verticillioides* and in the South *A. flavus* typically occur. From China, we retrieved only one source mentioning *F. graminearum*, *F. verticillioides*, and *F. proliferatum*, which were decisively observed in Southwest China; NIV and DON were also reported in *F. meridionale*- and *F. asiaticum*-infected regions [64].

The most dangerous two *Fusarium* species are *F. graminearum Schwabe* (*Gibberella zeae Petch.*) and *F. verticillioides* (*G. fujikuroi*). *F. graminearum* is a highly pathogenic species, which can cause significant yield quality losses due to toxin contamination with DON and zearalenon (ZEN). Meanwhile, *F. verticillioides* is a weak pathogen, with its significance lying in the production of a large number of fumonisin mycotoxins [65,66,67,68]. In Croatia, besides these two species, *F. proliferatum*, *F. sporotrichioides*, and *F. solani* have been isolated. In the rather dry Iran, *F. verticillioides* and *F. proliferatum* dominated, with *F. acuminatum*, *F. scripi*, *F. equiseti*, *F. semitectum*, *F. nygamai*, and *F. culmorum* also occurring at a low frequency [69]. Argentinian data indicated the dominance of *F. verticillioides* and *F. subglutinans*; and, less frequently, *F. graminearum*, *F. proliferatum*, and *F. cerealis* [70]. In Switzerland, Dorn et al. [71] identified 16 *Fusarium* species from grains, with dominant species including *F. verticillioides*, *F. graminearum*, and *F. crookwellense* in the north; and *F. verticillioides*, *F. subglutinans*, *F. proliferatum*, and *F. graminearum* (only 1.5%) in the south. They also observed 15 species from plant stems, with *F. equiseti*, *F. verticillioides*, *F. graminearum*, *F. crookwellense*, and *F. subglutinans* dominating. A global picture was given with similar data by Logrieco et al. [72].

Various *Fusarium* species, along with the toxins that they produce, are listed in Table 3. It is an important lesson for plant breeding that various toxins can be synthetized by different *Fusarium* spp.; the fungi cannot be identified solely based on toxin data. Therefore, it is not clear which species should be taken into consideration when breeding for resistance. On the other hand, when the fungus is known, in most cases, more than one toxin should be measured for the determination of food safety. The data speak for the introduction of the multi-toxin control, as, by measuring only one toxin, the food safety will be jeopardized.

The correlation between natural infection and toxin contamination is often not significant, and there were cases when toxin contamination was measured even though no physical symptoms were observed. We should mention that the members of the *F. graminearum* clade comprise close to twenty distinct species. *F. meridionale* and *F. asiaticum* have been observed to occur in China, and the 16 *F. meridionale* and 3 *F. asiaticum* isolates produced NIV, DON, and ZEN; furthermore, all *F. verticillioides* and *F. proliferatum* isolates produced the FB_1_ toxin [64]. In maize, data on their occurrence are scarce, and relevant studies generally did not mention resistance and toxin problems together. As wheat and maize often change in the crop sequence, their common pathogens have continuous propagation chance. At a higher resistance level and with better agronomy (good agronomic practice), the chance of an epidemic is smaller. In Brazil, *F. moniliforme* (now renamed *F. verticillioides*; accounting for 61% + 9% *F. subglutinans*), *A. flavus* (15%), and, surprisingly, *Penicillium* spp. have been shown to be the most dominant species. The *Penicillium* spp. are important, as they show the storage origin of Brazil samples [74]. This is similar to the case for *A. flavus*, which requires further research. There is another problem that influences the information value of the data: the resistance levels of hybrids remain unknown. At a higher resistance level, a lower infection level is probable. As highly susceptible hybrids give a higher rate of the isolates, this modifies the picture. As the resistance levels of the hybrids to different toxigenic species are mostly unknown, the species composition will not necessarily agree with the occurrence of the different toxigenic species in the field or region calculated only according to meteorological and environmental data.

The species composition is not stable [75] in Hungary; for example, *F. verticillioides* was present in 17% and 65% of isolates tested in 1974 and 1975, respectively, while *F. graminearum* was stable at 28% and 29%, *F. oxysporum* was present in 7.76% and 0.31% of isolates, *F. culmorum* was present in 4.33% and 0.47% of isolates, *F. fusarioides* was present in 3.12% and 0.62% of isolates, *F. sporotrichioides* was present in 5.95% and 0.16% of isolates, *F. poae* was present in 1.62% and 0.16% of isolates, and *F. semitectum* was present in 3.61% and 0% of isolates, respectively. This means that, between years, differences in humidity and temperature can lead to very intensive changes. Therefore, the structure of the maize *Fusarium* population is typically unstable. In the wet year of 1974, the number of species isolated from one location ranged between 3 and 12 (23 locations), with a mean of 6.3 and a total of 16 species identified in the country. In the drier year of 1975, the number of isolated species ranged between 1 and 5, with a mean of 2.2 and 11 species in total identified in the country. This means that, in dryer years, less mycotoxins may occur. In years that have more moisture, however, more toxins at a higher concentration can be expected, as has been shown by Mesterhazy et al. [51].

We should mention that disease-causing ability and toxin contamination cannot be considered to be interchangeable. Although it is well-known that pathogenicity and toxin production may correlate for *F. graminearum* [76], this is not necessarily true for *F. verticillioides*, and the relevant relationships for other *Fusarium* species remain unknown. The hemibiotrophic character of *F. graminearum* has been recorded [77,78] in wheat; however, in maize, related information could not be found. For *F. verticillioides* in maize, supporting data are unavailable; however, such a relationship might be possible. The significance of this finding is that hemibiotrophic pathogens have a similar invasive ability to infect plants than rusts, several days later shifting to the more useful necrotrophic lifestyle. This is an explosive combination, which should determine the direction and success of breeding efforts.

The Aspergilli have been considered post-harvest pathogens for around 60 years. As described by Christensen and Kaufmann [79], they can grow in conditions of 70–90% relative humidity. A total of 3000 grains originating from the field were placed after surface sterilization on media, and *A. glaucus* was observed in only 4.88% of the isolates. At present, the most significant species for maize include *A. flavus*, *A*. *parasiticus*, and *A. glaucus*. Most of the relevant literature has dealt with *A. flavus*, while the other species have received little attention. Another test [79] in freshly harvested grains showed 48–83% *Alternaria* spp., 1.4–10.9% *A. glaucus*, 0–1.8% *A. flavus*, and 0–0.8% *Penicillium* spp. While other field fungi cannot survive lower than 12–13% grain moisture content and temperatures above 21 °C, the fungi detected in storage live also in the field and may be isolated at a lower or higher rate; therefore, their field origin cannot be questioned, especially when the storage facilities have been properly disinfected. As both *Penicillia* and *Aspergilli* have incredible spore production capacity, when the storage conditions favor their survival, their very rapid spread and microbial damage may result.

Varga et al. [80] tested well-stored and badly stored moldy maize samples and compared their mycotoxin contamination. Under well-managed storage, the mycotoxin contamination for the four field-originated mycotoxins (reprinted in [51]) was low, while the moldy samples contained many-fold-higher so-called “field toxins” that had actually been synthetized in the poorly managed storage rooms. For Aspergillus, the story is perfectly different. In the USA, Shannon et al. [81] reported the occurrence of aflatoxin at harvest in maize for the years 1964 and 1965. Anderson et al. [82] observed up to 400 mg/kg of aflatoxin in infected maize grains. These findings have also been supported by other authors [83,84,85]. This proof is important, in that solving preharvest toxin contamination by breeding for resistance alone or in combination with other agronomic practices is essential. This is also a problem when regarding field-originated mycotoxin contamination. It is not accidental that *A. flavus* was described to cause ear rot among the field-originated ear rot diseases as a primary classification [3]. In addition, its storage rot-causing ability is also valid. While Penicillium ear rots have also been recognized in maize ears before harvest, the main damage is caused during storage [3].

Maize is attacked by many *Fusarium* spp., and the total number may range above 20 of the members of the different *Fusarium* species complexes and clades. Breeding against natural infection by presupposing a complex resistance to different *Fusarium* and *Aspergillus* spp. is a false and outdated idea. This idea has not been formally stated, but there is a general conviction behind the idea that resistance to natural infection will resolve this problem. When we consider the high variation in the *Fusarium* spp. population from year to year and from location to location, as was observed in Hungary [75], we determined that a maize hybrid in a certain location will have a similar resistance to another population against the local fungi. No data support this idea. However, practical breeding has followed this idea for decades. For example, when the devastating *F. graminearum* epidemics in Hungary became less pronounced, resistance to natural infection was considered to have been achieved. As later epidemics verified, strong epidemics could not be prevented in this way. Relevant knowledge also increased, allowing for new alternatives to be provided through further research. We do not possess resistance data for most toxigenic fungi, and the relationships between disease resistance and toxin contamination are practically unknown. Thus, obtaining a full solution to this issue is not likely at present. First, the scientific evidence should be analyzed to determine all possibilities. The decisive question is whether we have a general resistance that is valid for these pathogens [86,87,88], or if we should breed separately with respect to the most important species and later pool the resistance in plants.

## 4. Which Mycotoxins Are Important?

Four considerations can be made in regard to the importance of the various mycotoxins.

The first is that the mycotoxins produced by the toxigenic fungi that occur most frequently in a given region should be listed first. Examples include DON and ZEN from *F. graminearum*, FB_1_ from *F. verticillioides*, and aflatoxin B1 from *Aspergillus flavus*. These mycotoxins have global significance [3]. To these, we should add the different masked variants, isomers, and so on. Then, locally important toxigenic fungi should be considered; however, breeding against these is normally not carried out, and resistance to the most important species remains unknown, so they are problematic and uncontrolled. It is not an accident that the brochures for hybrids normally do not address this problem, with the yielding ability being the most decisive element. As an example, members of the *Gibberella fujikuroi* clade, such as *F. verticillioides*, are known to produce MON (without exception) and BEA (rather rarely), causing additional food and feed safety problems [89].The second source of knowledge is multi-toxin analyses of infected food and feed samples. There are two modes of analysis: targeted and non-targeted. To obtain a full picture, a non-targeted analysis should be conducted, through which even 50–100 toxigenic compounds can be detected—mostly for research purposes—with the objective of determining those that should be the subject of targeted analyses. The authors of [89,90,91] identified 12 mycotoxins in maize and its products, with *Fusarium* toxins found mostly in whole maize and maize gluten samples. Aflatoxins, ochratoxin A, T-2 toxin, and HT-2 were concentrated in maize gluten feed. Meanwhile, the starch part was much less contaminated. Guan et al. [2] checked 11 mycotoxins, and, beyond the regular AFs, DON, FBs, and ZEN, OTA was also identified. Eckhard et al. [92] observed DON in every silo maize sample, but (in decreasing order) nivalenol, T-2, HT-2, acetylated DON, and FBs were also detected. From these studies, we can conclude that, beyond the standard four toxins, we should also consider nivalenol, T-2, HT-2, and (maybe) OTA; the necessary list may be much longer. Non-regulated mycotoxins represent an important analysis object, and, in one study, 38 different combinations were detected, with combinations in one sample containing up to 12 mycotoxins [93]. The extent to which this is a breeding problem remains unknown, but the problem is considerable. Among these mycotoxins, nivalenol (NIV), enniatin B (ENB), and enniatin B1 (ENB1) were the most frequent. As nivalenol can be produced by strains of *F. graminearum sensu stricto* and some *F. culmorum* isolates—as well as by *F. asiaticum* and other members of the *F. graminearum* clade—we should pay attention to such species.The third source of information is urine and blood mycotoxin analyses in humans and animals. Lemming et al. [94] tested 3000 school students for urine and blood sera. In urine, DON, DON-15-β-D-O-glucuronide (DON-15GlcA, 9.1%), dihydro-citrinone (DH-CIT, 0.5%), HT-2-glucuronide (HT-2-3-GlcA, 0.1%), and ochratoxin A (OTA, 0.1%) were identified, while OTA was found in all sera samples. Furthermore, 2′R-OTA occurred in 8.3%, and enniatin B in 99.2% of all samples. The cereal background was found for DON, ENB, and OTA. Izzo et al. [95] compared targeted (for six toxins) and non-targeted analyses on the same samples. They concluded that the targeted test was better when the toxin background had been determined prior. Abia et al. [96] identified aflatoxin M1, fumonisin B1, ochratoxin A, and DON from urine. De Santis et al. [97] screened eight mycotoxins from urine and found DON, ZEN, FBs, T2/HT2, and NIV to be at critical levels. Schmidt et al. [97] reported the occurrence of aflatoxin M_1_ (AFM_1_), altenuene (ALT), alternariol monomethyl ether (AME), alternariol (AOH), citrinin (CIT) and its metabolite dihydrocitrinone (DH-CIT), fumonisin B_1_ (FB_1_), and ochratoxin A (OTA). Additionally, ZEN, as well as α- and β-zearalenol, FB1, and OTA, was found in 100% and 38% of human urine samples, while AFM1, ZEN, citrinin, and DH-CIT were observed in 10–20% of samples. Novel Spanish data [98] reported on the presence of all aflatoxins, fumonisin B1 + B2, ochratoxin A, DON (with 3 and 15 ADON), and T-2-HT2. The overall list is longer, but at least 4–6 mycotoxins appear to occur at levels higher than the limits and, so, can be considered responsible (alone or in combination) for toxicoses in humans and animals.The fourth information source is the toxin regulations that set binding or suggested limits. The limits are binding for human consumption, while, for animal husbandry, suggested limits are provided for all except aflatoxins, as animals typically cannot be supplied with human-quality feed in agriculture. This is changing: as new research results on the critical toxicity of mycotoxins are being published and confirmed, the EFSA in Europe has suggested that the EU modify the relevant regulations [99,100,101]. Although the regulations do not have the same limits in all countries, human health and feed export/import considerations are controlled. For example, when an individual wishes to export maize from Europe, they must further consider the European regulations in the case when their home country regulatory limit is less strict. A recent review has mentioned mycotoxins such as trichothecenes, fumonisins, zearalenone, aflatoxins, and ochratoxin A as the most-researched items [102]; however, useful data for breeding are available only for DON, fumonisins, and (to a lesser extent) aflatoxins, and they are far from perfect. Other mostly emerging mycotoxins, such as fusarenon-X, beauvericin, enniatins (A and B), and moniliformin, have been obtained from both raw and processed cereals and possible resistances are yet unknown.

When we consider the present situation, in which most breeding programs concentrate on a single *Fusarium* sp. or *Aspergillus flavus*, the most important task is to understand the epidemiological situation in a particular region. As we are gaining more and more data on the occurrence of multiple mycotoxins, it becomes clearer that resistance to one or two toxigenic species may not be sufficient, and we must evaluate further solutions to control the presence of multi-toxins as much as possible. As stated above, all three main pathogens (alone or combined) caused epidemics in 2–4 years of the last 9 years in Hungary. For this reason, in Hungary, resistance to all of the main toxigenic fungi is necessary [51]. There are many such areas in the world where the situation is similar or better/worse. Normally, an epidemic reveal whether we are well prepared or will be less successful in regard to future events. As the emerging mycotoxins are now with us, and some of them will gain regulatory limiting values sooner or later, the problems associated with maize production can be expected to increase. The possibility of resistance development is a key task, but present knowledge is insufficient to bring the key species under control. New innovative technologies should also be developed and combined, which may help to avoid severe (i.e., double toxin content above limit) toxin contamination and prepare for their application in the field. *It is the author’s opinion that the cheap multi-toxin technologies available at present can support this work* (e.g., the MycoFoss multi-toxin analyzer, among others).

## 5. Resistance to Ear Rots and Toxin Contamination

### 5.1. General Considerations

Numerous papers have been published in the resistance field; however, in the maize genetic conference series, only several papers were published in this subject. In 2005, none of the 32 talks and only 3 of the 248 posters were connected to toxigenic fungi [103]. In 2006, 258 posters and 37 talks were given, of which 2 mentioned Fusarium and 5 aflatoxin [104]. In 2008, the abstracts of 54 talks and 241 posters contained no paper considering Fusarium or aflatoxin [105]. In 2022, out of 27 talks and 219 posters, 2 focused on Fusarium and none considered aflatoxin [106]. Although not all conferences are cited here, these few clearly demonstrate that the significance of toxigenic fungi is fairly low in the genetic field. The real situation is possibly not so bad. Moretti et al. [107] analyzed the relationships between climate change (mostly warming) and mycotoxin-related problems, forecasting the increase in aflatoxin in South, North, and Middle Europe. In terms of Fusarium mycotoxins, fumonisins and DON are expected to be the most prominent, but plant breeding for resistance and better adaptation were not mentioned among the most important practices to balance the harmful effects of climate change on food safety. The study of Dovenyi-Nagy et al. [108] is an exception, as they discussed the roles of resistance breeding in terms of reducing aflatoxin contamination in the field. Tirado et al. [109], in their review, mentioned mycotoxins in one line, and, as an initiative, only worldwide mycotoxin regulations were mentioned in terms of helping to control food safety standards; however, this cannot solve the problem. Other authors [110,111] have responded similarly.

Resistance can be grouped into the physiological and physical (morphological) resistance of the maize grains [8]. The literature references the influence of physical factors, such as husk leaves, ear angle, kernel characteristics, dehydration rate, and others [112]; general experience suggests that these traits alone can lower the risk but are not capable of solving the overall problem. More importantly, for example, the thickness of the seedcoat and its waxiness, which are genetically determined, can have a significant effect in protecting grains against FER infection [113,114]; however, what is the relative case for *F. verticillioides* and *A. flavus*? The husk leaves play two key roles. When they remain closed until ripening, the higher humidity under the husk leaves can support the spreading of fungi. Thus, when the husk leaves open and dry-down can be accelerated, this indirectly inhibits fungal growth by decreasing the water activity (a_w_) and, thus, the chance of infection. However, a vertically standing ear with opened husk leaves can lead rain into the ear, which can significantly increase the water content of the grains and support increased toxin contamination. The husk leaves on bowing ears protect the grains like an umbrella, and the water uptake of the grains remains moderate or close to zero. The dehydration rate of the kernel (drydown) also has an influence on ear rot and mycotoxin accumulation; for a long time, this has been considered an important breeding task to lower the dehydration cost after harvest. The largest problem seems to be related to the lack of new findings regarding the significance of physical barriers. Without any facts, we cannot hope to make any significant progress. It is understood that physiological dry-down and pathological dry-down following stalk rot inhibit fungal spreading [115]. The latter causes a pseudoresistance to ear rot that, in a rainy season, will not be effective and supports epidemics [115,116]. For this reason, the development of stalk rot should be inhibited, as ear rot—particularly that caused by *F. graminearum*—can give genetically valid results when stalk rot is absent or, on the scale of 0–5, only a rating of 1 is acceptable. This can be achieved through lower stand density, irrigation, or a combination of both. For this reason, in our tests, stalk rot is always checked. For ear rots caused by the much less pathogenic *F. verticillioides* and *A. flavus*, no data could be found, possibly because their influence on the grain yield is low [15]. The author would like to see more papers in this field, as it is not an accident that a vast majority of publications have utilized artificial inoculation methods. The argument that artificial inoculation inhibits the effect of the physical protection is not well founded, as toothpick or silk channel inoculation will not lead to spreading in a resistant plant, while 70–90% infection severity can be achieved in a highly susceptible plant. Therefore, the identification of the resistance is possible, allowing for the selection of the highly resistant hybrids [10]. It can be supposed that, at a high resistance level, the influence of physical resistance or susceptibility is much lower. As physical traits also have a genetic background, as much so as their physiological background, the differences between them should not be stressed but, rather, their combined application is suggested (when supported by solid scientific evidence).

The term “physiological resistance” could be changed, but its use should be avoided; for example, the waxiness of the seedcoat is produced by different genetic and physiological processes influencing plant behavior. The latest genetic research has described the resistance to toxigenic fungi in cereals as quantitatively inherited resistance (QDR)—meaning polygenetically determined traits [117], as, until now, no evidence has shown monogenic traits to secure high levels of resistance alone. It should be noted that most of the resistance genes again rust diseases identified in wheat determine only a moderate susceptibility or resistance level, and only the most effective can provide field immunity [118]. Furthermore, race nonspecific resistances governed by QTLs have also been determined with regard to rust disease in wheat [119]. Considering the two resistance types, the common factor is that both can determine partial resistance resembling QDR [117].

Munkvold [120,121] has summarized research from the USA and stated that Fusarium ear rot has not been reported in the literature, as found for *Gibberella zeae* and *A. flavus*. Since that the situation changed, the number of FER publications surpasses GER papers and is comparable with that of *A. flavus*. Hung et al. [122] added that all significant breeding activities were concentrated into the private industry, and, while the genetic basis was better explained, screening methods were found to be less adequate. This was the first summary considering QTL identification, and the polygenic nature of resistance was well supported. However, at this time, resistance to disease and toxin contamination was not mentioned as a separate problem. Munkvold [123] stressed the need for the appropriate selection of hybrids and found it highly important to understand the pathogen composition in a region, as the resistance relations and their validity may change outside the region. To date, the knowledge on FER has improved significantly, but the aflatoxin contamination has become more problematic, partly due to the warmer and dryer seasons. In addition, the lack of reliable data on resistance greatly inhibits both production and research.

Not accidentally, researchers have observed rather good correlations between the amount of the disease (FER) and the toxin contamination in several tests [124]. Functional genomics may help to identify further resistance paths against *F. verticillioides* [30]. Santiago et al. [29] summarized that the symptom severity in resistance tests correlates well with lower toxin contamination; however, they stated that direct screening for lower mycotoxin contamination is expensive and time-consuming. A key problem in this context is that the data typically come from experiments on inbreds [17,32,33]. For this, we need combining ability for grain yield and disease, as well as resistance to toxin accumulation that normally does not exist. For this reason, only hybrid resistance tests can advise us correctly.

It is widely accepted that resistance level and toxin contamination are closely related; thus, it is sufficient to select for high resistance to achieve low toxin contamination [29]. Eller et al. [125] stated that direct selection for toxin contamination is very expensive and, therefore, is not suggested. The highest infection was found six days after silking [48]. A key problem is that, without ear rot-severity data, the toxicity data cannot be validated. We previously detected a more than ten-fold difference in mycotoxin contamination for 1% of FER, GER, and AER infection severity [51] (Table 4), but it is not possible to forecast toxicity for a significant number of genotypes. As we do not know which hybrid will react in a certain way, toxin measurement is necessary. An important aspect of toxin production is that the percentage of visual infection may present very large differences. For a natural infection, the case is similar [10] for a percentage of ear infection, the DON content ranged between 0 mg/kg and 7.63 mg/kg for 2017/2018 and 0 mg/kg and 46.5 mg/kg for 2019/2020. The FUM content ranged between 0 mg/kg and 6.31 mg/kg for 2017/2018, where 1.2 and 58.2 mg/kg were the minimum and maximum rates, respectively. Aflatoxin ranged between 1.10 mg/kg and 63.5 mg/kg for 2017/2018 and 0–6439 mg/kg for 2019/2020 [10]. These differences revealed no significant correlations. The case is similar when conducting artificial inoculation (Table 4). The rate of maximum and minimum values was the lowest for DON (8.63-fold), medium for fumonisins (17.61-fold), and highest for aflatoxin B1 (58-fold). This serves as an argument to not predict toxin contamination based on visual infection.

Looking at the variances (this serves as a stability index) with regard to different toxins, large differences between hybrids were found [51]. Several gave a very uniform performance, and some others reacted very differently in the two years to the two isolates. So, hybrids with stable performance to different toxins and their producers could be identified. The summary table (Table 5) shows all data of the hybrids that are important for the risk analysis [51]. Among them, four hybrids were identified for which all of the data were in the low and low–medium groups. Only one hybrid had the worst (orange) category for all toxins, while the rest varied. So, without any logical order (e.g., from the DON response), no conclusions could be drawn for the other hybrids. Of course, the original data were also important in performing a risk analysis, as they reveal the causes of the correlation-breaking genotypes; however, even the specific DON-regulating agents remained unclear. Regardless, a risk analysis can still be conducted. On the other hand, such research can help to clarify the causes of the phenomenon.

In both research and variety registration, toxin analyses cannot be spared. Through the breeding process, we have the possibility to reduce the number of toxin tests. When a hybrid is susceptible, it can be discarded without a toxin test. Most importantly, those hybrids that have good resistance, but high toxin contamination are dangerous. For example, in 2017/2018, Korimbos and DKC 4590 had 0.04% and 0.36% FER severity at 0.81 and 2.31 mg/kg FUM, 0.05% and 0.15% AER severity at 298 and 2150 mg/kg AFB1, and 8.48% and 11.18% for GER with 26 and 9 mg/kg DON. At the same time, the naturally infected controls of several hybrids showed no visible fungal presence for AFB1 and presented 32 mg/kg contamination. Similar examples were also observed in the 2019/2020 hybrid tests, and, under natural infection, several hybrids produced 350–450 mg/kg aflatoxin or higher without visible infection. The extent to which such results are environmentally dependent remains to be determined. We concluded that the associated relationships are much more complicated. Even the resistance and toxin data correlations were variable for the different ear rots; they allow us to identify superior genotypes for good performance to all diseases and the accumulation of toxins [10,50,52,126].

### 5.2. Gibberella Ear Rot

GER is favored by higher precipitation during silking, followed by moderate temperatures, and high rainfall in maturing period, as occurred in 2014 in Hungary [10,120]. The causing agent, *F. graminearum*, is highly pathogenic. Its isolates are mostly highly aggressive; therefore, it is easier to work with in a breeding program than the much less pathogenic *F. verticillioides* or the even less pathogenic *A. flavus* [51,52]. The susceptibility window is about 6 days during silking [44,45]. Miller et al. [127] presented data on the process of silk infection. The cob was accessed in 5–7 days in susceptible genotypes but required 12–15 days in resistant genotypes. Entry into the ear was through the surface of the rachis, via the exterior growth of the grains or in the rachis through the pedicel. This would mean that inoculation is ideal at between 5 and 7 days after mid-silking, allowing for better a differentiation of the resistance level. The subjects of the tests were very early hybrids, as they are in commercial production, and their resistance and toxin risks have direct economic consequences [47]. They pointed out that less than 10% of the tested hybrids presented a stable performance. Our recent results [50,51,52] fully support this conclusion. The inheritance of the GER resistance had the highest rate for GER symptoms (17% increase across 27 hybrids). For DON, the mean heterosis was −35%; e.g., the majority of the hybrids had a higher DON content than the mean of the parents, and only seven hybrids were better than the parental mean was. So, the heterosis in symptoms will not agree with the DON contamination of the hybrids; the case seems to be more complicated than supposed [126].

QTL research. Ali et al. [40] identified 11 QTLs for *F. graminearum* severity for ear rot resistance following silk inoculation, as well as 18 QTLs in kernel resistance, which accounted for 6.75–35% of the total variation identified. Only two QTLs were detected in more than two environments for silk resistance, and only one for kernel resistance. The disease evaluation followed Reid et al. [44,128]. The authors explained this result in terms of the high influence of the environment on the expression of resistance. In the author’s opinion, this might be true; however, the facts that all QTLs had a small effect and that the total variation (even in the best case) was only one-third of the total variation make it likely that there may have been problems in the testing methodology. The 1–2 QTLs that were active in all environments could make a large improvement in the resistance when used, while controlling only a small percentage of the total variation. In the test, no toxin contamination assessment was conducted, and, so, it is not known how far these resistance results may contribute to lower mycotoxin (e.g., DON and ZEN) contamination. Four QTLs were identified by Galiano-Carnerio et al. [42], explaining 5.4–21.8% of the genotypic variance. Of these, only one was stable across all environments. Toxin data were not included.

Giorni et al. [129] identified four QTLs in the LP4637/L4674 population (No. 298 RIL lines), explaining 11.2–11.8%, 3.4–5.1%, 6.2–7.6%, and 3.8–5% of the total variance (24.6–29.5%). This may be consistent with other sources, but the effectiveness of a breeding program that can maintain only 25–30% of the resistance is questionable, as this means that we do not have an explanation for 70–75% of the variance observed. The validation of QTLs is a general problem: out of five identified QTLs, none was stable, and the validation rates were disappointingly low, indicating GER resistance arising from many low-effect QTLs [130]. A total of 140 RIL genotypes have were tested [43] for silk and kernel resistance. These RILs showed highly significant differences. The QTLs were co-localized in chromosomes 1, 2, and 8 and differed in chromosomes 9 and 10. Chr. 9 seems to have an influence on silk resistance, while Chr. 10 seems to be specific to kernel resistance. As the data did not present any correlation with agronomic data, the grain yield response seems to be independent (only weak correlations) from GER resistance. As the disease severity did not differ significantly, the differences were not derived from the differences in aggressiveness between the two inoculation methods. Zhou et al. [46] identified 11 QTLs, 5 of which were stable. The inheritance was additive + epistatic, and no one QTL explained more than 10% of the phenotypic variation; furthermore, no toxin measurements were conducted.

Yuan et al. [131] were among the first to identify a putative resistance gene (guanylyl cyclase-like gene) against *F. graminearum* in maize. Reid et al. [44] tested the correlation between visual scores and DON contamination in hybrids. Ear rots higher than 25% GER contained high levels of DON. They concluded that visual symptoms should be suitable for breeding resistance to lower toxin accumulation and that no specific toxin measurements are necessary; this can be valid for this test, as we have cases for good agreement and no significant correlation, too. The problem is that high DON contamination can also be detected at 10% infection severity or lower. Tests with updated methodologies and toxin sampling showed that, depending on the experiment, 20–30% of the hybrids reacted differently, presenting higher (DON overproduction) or lower (relative DON resistance) DON contamination than that forecast by the regression function [10,48,52]. This holds also for the natural infection, but the correlations were even lower and mostly not significant [51]. We found hybrids presenting 25.8 mg/kg DON at 8.58% GER and 10.84 mg/kg DON at 12.33% GER in 2017/2018. Among the genotypes for 2019/2020 in the same paper, we observed a hybrid with 13.50% GER and 7.48 mg/kg DON, and another with 14.35% GER and 76.08 mg/kg DON. Similar deviations were found in a combining ability test of maize inbreds for GER and DON contamination, and the results indicated a more complicated situation than supposed by the literature result cited above [126]. The GER and DON presented a correlation of only r = 0.54. For the hybrids, 19 reacted similarly with regard to GER and DON, while 8 gave very variable responses, indicating a much more complicated inheritance mechanism. Another study [132] identified 4–6 QTLs for GER resistance and resistance to toxin accumulation in a mapping population of two inbreds, which explained 29–35% of the phenotypic variance. While the authors [132] supposed a pleiotropic effect, the explanation may be different: when resistance and toxin contamination are closely correlated, it means that a lower infection severity automatically means lower DON contamination or higher severity with higher DON contamination, without any pleiotropic effect. When we tested hybrids with highly different origins, the picture differed, and a larger range of variation was observed [51,52]. Genome-wide association studies and genomic selection have also been conducted in order to utilize the higher resistance found often in landraces. Considering GER severity, eight QTLs (explaining 34% of the genetic variation) were identified in the Kemater Landrace Gelb which presented no significant correlations with agronomic traits such as days to silking and seed set [60]; this is good news in terms of breeding for higher grain yield in all ripening classes.

Other approaches might also be interesting. Geranic acid has been found to act as an antifungal agent against *F. graminearum* in maize. The authors produced transgenic maize plants via geraniol synthase. They observed a geraniol dihexose and four different hydroxyl-geranic acid-hexoses in the transgenic plants; however, these plants were not more resistant than the control material [133]. A maize defensin peptide, PDC1, which is expressed in *Escherichia coli*, has been assessed in maize against GER. The peptide restricted fungal growth and disease severity, but the influence on toxin content was not investigated [134].

It seems that resistance to GER is more complicated than previously thought. The generally accepted excellent correlation between visual symptoms and DON contamination is not generally valid, with a smaller or higher part of the tested population not fitting the assumptions. For this reason, DON measurements cannot be excluded. No toxin measurements were made in many studies; therefore, the food safety risk of the hybrids cannot be reliably evaluated. Food safety problems are caused not by DON but, instead, its masked and acetylated versions, which are normally not controlled. Zearalenone data are nearly absent; however, it can be considered important to understand how resistance to DON accumulation corresponds to ZEN data. We should mention nivalenol, which has been found to be produced by isolates of *F. graminearum*, *F. culmorum*, *F. asiaticum*, and other *Fusarium* spp. As it is ten-fold more toxic than DON, its food safety risk may be highly significant. QTL analyses allow for one certain conclusion: the trait is polygenic, and the explained variability is seldom higher than 30%. This means that methodical problems may also contribute to this phenomenon, as evidenced in wheat [135,136].

### 5.3. Fusarium Ear Rot

For FER, the optimum temperature is 30 °C. In order to develop, it needs warm and dry weather during grain filling. Drought stress increases disease severity. The silking period also provides higher disease incidence and severity [120,137]. Significant resistance differences to FER have been documented in a long list of papers. In the test inbreds, hybrids were tested per se, but genetic studies between susceptible and resistant inbreds have been conducted using many different methods. Inbreds are generally more susceptible than hybrids and show highly significant resistance differences. In 2009, 0.56 and 240 mg/kg of fumonisin were measured by Balconi et al. [138], but the correlations between symptom severity and fumonisin content were only r = 0.37 and r = 0.29. On this basis, the prediction of fumonisin production based on symptom severity does not seem to be very promising. Significant differences have also been found for hybrids [10,47,50,52]. Fumonisin production co-segregated with the aggressiveness of *F. verticillioides* isolates, as was observed for DON in GER infection [139]. As the aggressiveness of *F. verticillioides* isolates is generally low, preselection of the isolates for artificial inoculation is necessary [50,52]. From a methodological point of view, it is interesting to note that lower and higher FB1 + 2 concentrations were found in high- and medium-rainfall zones in Zambia, respectively, without correlations with years and precipitation. In South Brazil, this is the ruling species (Tessmann, pers. comm., 2023). The planting time was neutral. Consequently, high- and low-resistance hybrids were not found under natural infection; however, under artificial inoculation, significant differences between hybrids were identified [34].

To assess the resistance to FER, Araujo et al. [15] compared three inoculation methods: aspersion (spraying), injection, and natural infection. In two locations, the control and inoculated variants did not differ significantly while, in the two other locations, the artificial inoculation was stronger, but the results differed. Differentiation of resistance was better in the regions with a higher temperature. It seems that the female flowering and drying of the grains were the most sensitive phases in terms of infection and fumonisin synthesis [20,29]. For the latter, insect wounds were mostly found to be responsible. Therefore, in resistance tests, insect-damaged ears should be discarded. It seems that the dent stage is critical, and the highest fumonisin accumulation is when just reaching full ripening; here, no insect is needed. The role of husk coverage from earlier papers was supported here [140]. A non-excessive pericarp thickness also helped to reduce the fumonisin level. The structure of the pericarp and its wax layer can act as resistance factors protecting maize against *F. verticillioides* and fumonisin contamination, as removing the wax layer increased the infection and FUM contamination severity [113,114,141,142]; furthermore, the diferulate concentration appeared to be proportional to the fumonisin contamination (R^2^ = 0.82). Ivic et al. [28] could not prove this role in hybrid tests. Both may be true, and we observed this often when testing different hybrid groups [51,52]. Munkvold [120] reported on significant resistance differences with respect to FER, but the genotype responses differed in the network of countries and CIMMYT. Afolabi et al. [14] reported a significant correlation between the incidence of ear rot and fumonisin contamination (r = 0.39 and 0.35 in two locations; *p* = 0.0001), but they stressed that only five inbreds were found that produced consistently low fumonisin contamination (lower than 5 mg/kg) across years and locations. This agrees well with our data [10,50,52]. These inbreds were successfully used in a breeding program. *This is important*, *as these authors were among the first who stressed the significance of a stable low toxin response*, and their practical significance could also be verified.

The combining ability is an important trait, and additive and non-additive gene effects should be identified [143]. Hung and Holland [122], in a diallel analysis, observed a 27% decrease in infection severity and 30% reduction in fumonisin content, compared to the parental inbreds. The general combining ability (GCA) and specific combining ability (SCA) were closely correlated (r = 0.78 or higher). Morales et al. [144] evaluated external and internal infection, as well as FUM contamination. The four families provided a large variability in response, and genotypes with lower kernel density and larger cobs had higher FUM. It seems that QTLs could be classified as putatively resistance-specific and putatively for ear and resistance traits. This result supports the different QTL functions observed in wheat [135,136]. This method is used mostly for grain yield breeding and is seldom used in resistance breeding. However, more recently, molecular methods are more commonly used, but the significance of the combining ability has remained important.

QTL analyses. Resistant and susceptible lines were used to create a mapping population, and three QTLs were detected on chromosomes 4, 5, and 10. Of these, Chr. 4 explained 18% of the 90% higher infection severity variation and, so, was selected for further work by Chen et al. [145]. No fumonisin tests were conducted. Wu et al. [35] analyzed FER resistance by linkage mapping and GWAM. A total of 10 QTLs were identified that explained 1.0–7.1% of the phenotypic variance. Epistatic mapping was better, which explained 21–30% of the phenotypic variance. As such, multiple genes were identified with minor effect. No toxin tests were conducted. Giomi et al. [20], through their QTL analysis, could explain 56–58.2% of the observed variation; this is among the highest that we found. Cao et al. [21], from 364 differentially expressed genes (DEGs), identified four QTL genomic regions against FER which were active with respect to disease and toxin contamination resistance. A positive development is that gene functions could also be connected to these results, such as those related to cell wall biosynthesis and flavonoid biosynthesis. This is important, as the possibility to identify genes can play a decisive role in resistance development and expression. Another possible gene group in this line is connected to the salicylic acid and steroid signaling system in silks [146]. Butron et al. [18] detected 13 putative QTLs with respect to FER, located on all chromosomes except Chr. 5, all of which were identified as small-effect QTLs. Of them, two seemed to have an influence on FER and fumonisin contamination when inoculation tests were conducted without replications (in each block of 42 genotypes, supported by eight lines from the MAGIC population as controls); their reliability should not be overestimated. The rest varied; for example, QTLs with different functions were found. Data were transformed using the log_10_(*p*) method. One should be cautious when drawing far-reaching conclusions from non-replicated trials. Perez-Brito et al. [147] could explain only 44% of the variability according to the QTLs identified; in retrospect, this is not a bad result, and most of the wheat QTL analyses have not provided significantly better results.

SNP and GWAS analyses. Genome-wide association mapping in 818 tropical inbred lines identified 45 SNPs and 15 haplotypes connected to FER resistance with 1–4% individual effect and of additive character. Altogether, five QTLs were identified in Population 1, which explained 49% of the total variance. For Population 2, six QTLs explained 25% of the phenotypic variance [22]. They concluded that the value of the identified QTLs was limited, indicating a polygenic inheritance which differs from population to population. They stated that a combination to select with general adaptation may help in increasing FER resistance. It should be noted that toxin contamination was not analyzed; therefore, the food safety aspect of the results is unclear. They applied the kernel resistance test (nail punch/sponge inoculation) described by Drepper and Refro [148]. Another study [36] revealed seven SNP variants across 556 hybrids associated with 1–3% of the phenotypic variation; however, FUM was not controlled. Santiago et al. [149] reported about 23 mapping populations on FER. Visual symptoms were analyzed in 19 populations only, and only 4 related visual symptoms to fumonisin contamination. This is very important, as breeders can better consider the fumonisin-producing capacity and differentiation of the breeding material. At this time (2020), no genes were known to underlie the QTLs. However, their data strongly supported the view that more effective breeding against FER is possible. Genome-wide association mapping (GWAS) is a new technology which can be conducted on hybrids and inbreds. As such, a large amount of material can be screened in a relatively short time. Coan et al. [150] identified 14 SNPs, among which four genes were identified as defense-related proteins. They explained 15–25% of the phenotypic variance, but none was stable across the three environments tested. Zila et al. [151] reported the GWAM results of 267 inbred lines in two environments. Eight SNP loci were identified with small additive effects, explaining 3–12% of the genotypic variation. Two SNPs were colocalized with genes having participation on programmed cell death. Another paper using GWA mapping identified FER QTLs and collocated the FER and FUM QTLs in five lines, and one determined only an anti-FUM effect. The high variation in the FUM data indicates that the additional genetic regulation of the FUM contamination is likely controlled by many other QTLs [24]. They stated that the chance of this is low, however, as the detected QTLs covered only a small percentage of the variation, and many remained undetected. For this reason, the detected fumonisin differences are considered less reliable than visual resistance data. For most QTLs, they detected a high level of instability across environments, with heritability ranging between 0.36 and 0.56 [24]. In maize cultivars, phenolic compounds presented high concentrations (822 mg/kg tyrosol equivalents) and had high resistance to fungal penetration and spreading by *F. verticillioides* [16]. Other resistance studies have identified resistance factors to FER such as the protein phosphatase 2A subunits [152].

Another approach is elicitor research influencing plant defense systems. Small et al. [153] tested the elicitor effect under known fungicide treatments; however, none of the elicitors and fungicides controlled FER and fumonisin contamination. The authors hope that better spraying technology, dose, and/or timing may lead to better results. On the other hand, this test proved that elicitors do not have general value, and we should identify specific elicitors (when they exist at all) in this context. Another paper went further and conducted a genome-wide association study on 183 inbred lines. Of 14 SNPs connected to FER resistance, 2 were connected to the starburst symptoms that seldom occur on FER-infected grains. Several QTLs were co-localized with known QTLs, and four were linked to SNPs encoding known defense-related proteins [150]. This is important, as it indicates that the QTLs slowly receive genetic functions. Another paper also considered SNPs and identified 28 QTLs; however, only the *qRcfv2* gene could be validated.

A new QTL was determined by quantitative PCR, but no expression related to resistance was verified [154]; furthermore, no toxin measurements were made. An interesting development is the identification of the *FvMK1* gene for *F. verticillioides* in maize, which is an ortholog of the genes *FMK1* in *F. oxysporum* and *GPMK1* in *F. graminearum* [155]. Importantly, the gene regulates multiple signaling pathways, pathogenicity, and FB1 biosynthesis. The question of how it behaves in resistant and susceptible maize genotypes remains open. Another aspect in this line of research [156] is that natural infection might have breeding aspects that have seldom been applied. A total of 98 inbreds from Lancaster, IDT, SSS, and SSS/IDT groups were compared, and kernel and cob fractions were also compared for toxin contamination. The visible symptoms and fumonisin content were significantly correlated. However, in some inbreds, the cob had higher fumonisin content than the grains. The explanation for this is not the pathotype of the pathogen but possibly the larger water content of the cob, such that the fungus can grow faster and inoculate the germ part of grains sitting on the rachis [52]. The difference between grain and cob moisture content varied between 11.5 and 28.3%. As fungal spread stops below 23% grain moisture [79], fungal growth is possible in the cob for 5–10 days (or more), such that the toxin contamination can increase freely. This is also resistance dependent. Maize varieties have been tested in two locations in Italy: Bergamo and Cremona. The FER response after artificial inoculation was rather divergent; however, two varieties (VA117 and VA1213) were identified that had stable resistance in both locations [31]. Another important aspect of relevant research was the identification of genes that play a role in FER resistance, such as the jasmonate-mediated monocot-specific 9-lipoxygenase *ZmLOX12* gene, which seems to play a key role in resistance to FER [157]. The gene also acted against FB_1_ accumulation. Disruption of the gene, however, allowed for the development of severe FER symptoms and high FB_1_ contamination. In one study, breeding for *F. verticillioides* resistance was successful, and five experimental hybrids were identified with low (<4 ppm) fumonisin level [23].

Naturally, genetics-based studies have attempted to identify resistance genes, including gene analogs or putative genes having an influence on resistance expression. Zhang et al. [155] identified the *FvMK1* mitogen-activated protein kinase as regulating conidium formation, pathogenesis, and FUM production, which seems to have some significance for plant infection.

The progress in basic research is clear, and the presence of genetically determined resistance to FER is proved and accepted. However, the toxin responses seem to be much more divergent than supposed. For this reason, as in *F. graminearum*, the key task is not determining the correlations and their closeness but the selection and identification of less susceptible and toxin-contaminated inbreds and hybrids. Both in the cited literature and according to our experience, 20–30% of the genotypes from inbred and hybrid testing may belong to this group. The problem of the explained variability is also a key factor. What can we expect from a QTL determining only 4% of the variation? The effect is probably not significant. In the author’s opinion, methodical problems may also be among the causes for these issues. By decreasing the background noise, better data can be obtained. Many candidate genes should be tested further, and the results may be surprising. To have a chance to make stable calculations, a rather broad database is necessary.

### 5.4. Aspergillus Ear Rot

At present, aflatoxin seems to be the largest problem of the three toxigenic species discussed. This is the least pathogenic, and, so, it is not easy to perform well-characterized experiments. About 50% of 80 isolates were found to be toxin-producing on rice, and only seven isolates were capable of producing toxins when inoculated into maize ears (Tóth 2023, unpublished). Therefore, the selection of a proper isolate requires care. Regarding this species, toxin data show high divergence and variability, and, in this case, it happens rather regularly that significant aflatoxin is produced without visual infection [51]. It is not an accident that the different methodologies for AER have the largest share, as aflatoxin is the most toxic of the mycotoxins that we normally work with. As resistance is not easy to obtain and aflatoxin often gives variable results, it often occurs that strategies are suggested to solve it without including resistance. Thus, several strategies may be combined to inhibit aflatoxin contamination or destruct it to produce a healthy crop free of aflatoxin, even if increased host resistance seems to be the best solution [4]. Many articles and reviews analyzing the management of the aflatoxin contamination do not speak about resistance as a possibility to combat the problem [158], at most mentioning the necessity of the more resistant varieties are necessary, without any suggestion of how to achieve this. However, the publication of resistance studies and results is increasing. Womack et al. (2020) [159] mapped 241 F_2:3_ families for aflatoxin contamination. Xu et al. [160] mentioned that resistance differences were presented but did not include information about registered hybrids against *A. flavus* and aflatoxin contamination. Kumar et al. [161] did not mention the possibility of utilizing plant resistance. Seed trade firms who suggest aflatoxin control plans for growers typically do not mention the use of more resistant hybrids [162].

On the other hand, there is nearly 50-year-old research presenting significant research results indicating that resistance and breeding for resistance provide a good perspective for the reduction of aflatoxin contamination. This is the reason why we take this research direction seriously and analyze the problems of how to manage it with a much higher efficacy.

It is seldom that an acceptable correlation exists between Aspergillus symptoms and aflatoxin contamination [10,51]. In several cases, visually symptomless genotypes could yield several 100 mg/kg of AFB1, and, in several genotypes, very high overproduction of AFB1 was the case compared to the symptoms. In other cases, AFB1 contamination was much lower than predicted according to the ear infection severity. All of this was observed as a mean reaction of two isolates separately applied in the tests. It also occurred that the correlation between the results of the two isolates was also seldom significant. As a result, in 2021, we used three independent isolates to identify *A. flavus* resistance and aflatoxin contamination with higher reliability. The main result was that we identified several hybrids with low variance across isolates and years for both infection severity (resistance) and low aflatoxin contamination. For others, extremely high variances were also computed. This agrees well with the conclusion of Wilcox et al. [163]. As the aflatoxin production for 1% of isolates was the largest for AER (57-fold), many of them were correlation-breaking; it is no wonder that a generally valid correlation seldom would be the case [51]. This underlines the necessity to identify hybrids with both low disease severity and low aflatoxin contamination, which was observed in about 20–30% of the tested hybrids; the rest may be suitable for the discovery of pathways for increasing or decreasing resistance-independent aflatoxin synthesis [51]. Wahl et al. [164] tested 295 experimental hybrids, in which the mean aflatoxin was 323–370 mg/kg. They identified 13 high-yielding hybrids with a lower aflatoxin level than the resistant control.

The polygenic nature of resistance and the low pathogenicity of the fungus do not help much to resolve the problem [51]. On the other hand, many papers have reported significant resistance differences between maize genotypes, supporting the reality of breeding for resistance [8,159]. Aspergillus ear rot and aflatoxin with fumonisin contamination together can follow damage by insects [3,63]; therefore, insecticides or *Bt* genes controlling FER and AER can inhibit toxin accumulation. The observation of preharvest aflatoxin contamination by *A. flavus* without insect damage indicated the infection ability of the fungus without any insect mediation and raised the need to gain resistance to the fungus and its toxins [85]. This is the subject of this treaty. In spite of the discovery of the significance of preharvest aflatoxin contamination in maize [83,84], *A. flavus* was thought to be a storage fungus, following the classic terms [79]. Therefore, in Europe, the preharvest nature of aflatoxin contamination was only proved recently; for example, in Hungary, this was in 2022 [51]. Based on this, the preharvest contamination was not controlled and was classified automatically as post-harvest contamination.

Earlier and recent findings cited in this review support the view [8,85] that real genetic differences exist, and resistance is believed to be primarily additive. It was stressed 36 years ago that both genetic and non-genetic management should be applied, as no practices had a major and consistent effect. It was added that resistance and aflatoxin regulation might arise from different mechanisms. Furthermore, the resistance of dent inbreds was about eight-fold better than that of sweetcorn inbreds. There are examples of studies on *A. flavus* resistance and resistance to aflatoxin contamination [108]. Abbas et al. [165] tested maize hybrids for natural aflatoxin and fumonisin contents. AFB1 varied between 27 and 641 μg/kg, while, for FB1, the variation was between 12.1 and 70 mg/kg. As the data came from one year, we can only suppose that genetic differences were behind these variations. As tests were conducted in freshly harvested maize, the mycotoxin contamination was of a preharvest character. It is remarkable that the aflatoxin contamination was, in epidemic years, comparable with Hungarian data at a very low natural visual infection level. Adejuma and Adejoro [90] posed genetic resistance as a possible solution, only citing previous data. Munkvold [121] reported on the early achievements in aflatoxin resistance research in the USA. Inbreds and hybrids with good resistance were identified and bred and were also introduced into the market. Active breeding programs have also been reported. Notably, it had been stressed 20 years ago that plants under stress are more susceptible to toxigenic fungi. Darrah and Zuber [166] have reported strong and significant differences in the aflatoxin response between flint and dent inbreds; the flints were generally better, but the results differed between locations. Henry et al. [38] chose to take a practical approach, identifying resistant inbreds (KO679Y and CUBA117:S15-101-001-B-B-B-B) through resistance testing, along with similar resistant and susceptible control lines with known response to *A. flavus*. The hybrids were promising in terms of both symptom and aflatoxin reduction. The disease pressure was high, so differences in resistance were well-differentiated. Henry et al. [27] inoculated 20 inbreds with *F. verticillioides* and *A. flavus*. For FER, the correlation between ear rot severity and fumonisin content was r = 0.74, while that for *A. flavus* was r = 0.61 (both significant). Of the 20 hybrids, 3 were resistant to both fungi and their toxins, 3 were highly susceptible to both, and the 14 others varied (the aflatoxin data seemed to be very high, such as 10,592 mg/kg (I checked the original data, this is written, but might be a mistake instead of μg/kg), even though it was ln(y + 1)-transformed data. However, they concentrated on the use of atoxic *A. flavus* strains to reduce aflatoxin contamination. The subjects of the resistance tests were mostly inbreds, but hybrids were also tested for practical reasons [164]. A recent review [167] identified stress-related genes and antifungal proteins in endosperm, rachis, embryo, and silk tissues before *A. flavus* infection and following inoculation. These proteins increased at a much higher speed in the susceptible lines. Plants having a transformed a-amylase inhibitor protein originating from *Lablab purpurea* could destroy 56% of aflatoxin contamination in the kernel tissue.

QTL, SNP, and GWAM. Aspergillus resistance in maize is still an unsolved problem. Baisakh et al. [168] concluded that, while AER-resistance loci could be identified, their phenotypic expression was very variable. In a meta-analysis of 276 from 356 identified QTLs, 58 MQTLs were identified in all 10 chromosomes. A meta-analysis of the differently expressed genes resulted in 591 genes with a putative response to *A. flavus* only. Of these, 14 SNPs were identified, and 12 MQTL-linked (meta-QTL) SSR markers identified three markers that could discriminate 14 resistant and 8 susceptible cultivars. However, no aflatoxin mapping was made, such that forecasting of the aflatoxin content could not be concluded. Another paper [169] reported that most commercial hybrids are susceptible to *A. flavus* and aflatoxin contamination. Through a multiple QTL analysis, several dozen QTLs with small effect were identified. An additional Pathway Association Study (PAST) highlighted several genes, including mechanisms that could determine a more or highly specific and well-inheritable trait allowing for more effective genomic selection. Aflatoxin data were not analyzed. It seems that the small-effect QTLs provide low probability to reach an effective genetic tool, even though a convincing resistance difference is demonstrated in Willcox et al. [163] verifying QTLs against *A. flavus* from the resistant maize donor Mp313E under different genetic backgrounds, which were stable under different environmental conditions. Of the 20 QTLs, 5 were identical to those mapped in the Mp313 × B73 mapping population. For this reason, we can determine valuable and stable sources for breeding; on the other hand, the 15 QTLs without stability should not be considered target QTLs. Another paper [170,171] conducted *A. flavus* inoculation, but only aflatoxin contamination was measured in order to supposedly identify the disease and toxin resistance. From the identified QTLs, only a minority proved to be effective for more than one year, and three of those detected in 1996/1997 were tested further for breeding use. As symptoms were not tested, the relationship between fungal resistance and resistance against toxin accumulation could not be verified. The other important conclusion was that the stability of the QTLs (or stability of the resistance) is a key problem in breeding. Another important result is that, in 1990, the Aspergillus-resistant germplasm Mp313E was released, followed by Mp420, Mp715, Mp717, Mp718, and MP719 [172,173]. The only problem here is that all aflatoxin data were transformed using the function ln(y + 1), such that the real aflatoxin data remained hidden, and it is not possible to see the original toxin concentrations. When grain is sold, the key question is whether the total aflatoxin contamination is lower or higher than 20 μg/kg. The ln(y + 1) function is uninteresting. As *A. flavus* data come mostly from temperate regions and not from the most-affected tropical region, tropical maize may become an important tool for finding better sources of resistance [174]. Seven SNPs have been found to be significantly correlated with AER resistance. A pathway analysis revealed 56 associated pathways, including phytoalexins, signaling hormones, and starch biosynthesis growth, as well as metabolic pathways. It seems that GWAS and pathway analysis can identify resistance candidate genes. The chitinase genes of maize might play a similar role [175], and one such gene is close to the peak of a recently published QTL [176].

Musungu et al. [177] detected an aflatoxin regulation cluster based on reactive oxygen species, which influences vesicular aflatoxin transport. As these events occur in the first 1–3 days after inoculation, this might explain early Aspergillus infection and aflatoxin transport and production [177]. Another innovative technology is gene silencing [178]; in this case, the *A. flavus* alkaline protease was tested, and eight variants were produced and backcrossed to susceptible parents. In the laboratory, an 84% reduction in aflatoxin was observed. The small RNAs presented a 1000-fold higher concentration in transformed plants; therefore, it is probable that the lower *A. flavus* severity and aflatoxin contamination is a result of gene silencing, achieved through the use of host-induced gene silencing (HIGS, host-induced gene silencing) technology. Another result in this line is the silencing of the kernel-specific RNA gene cassette targeting *aflC* gene [179]. No aflatoxin was observed in the transformed plant, while, in the control, aflatoxin at the level of thousands of parts per billion was recorded. Antifungal proteins have been identified [180] at higher levels in resistant plants compared with susceptible ones. A stress-related peroxiredoxin antioxidant (PER1) was cloned, and its expression in the resistant genotype was significantly higher than is a susceptible B73 line.

The question remains the same as Warburton and Williams [39] have asked previously: “Will this work lead to practical resistance results against *A. flavus* and aflatoxin contamination?” The presented data over several decades [83,84,85,166,181,182,183], along with later published papers (cited by Mesterhazy) [8] and more recent publications, have presented convincing data for differences in resistance; therefore, our answer is yes, this is possible. Nobody said that this resistance is immunity. It is polygenic and partial. The most complex polygenic trait is yielding ability. In spite of this, nobody states that a breeding for higher a yield would be impossible. Resistance testing methods have been evaluated, presenting convincing, significant, and large resistance differences that are suitable for utilization in breeding. Later developments concentrated on the molecular genetic work, but the underlying methodology remained the same. As the basic data were problematic, all conclusions had this problem. Statistical methods have even been developed to improve the quality of the data, with moderate success. At the same time, it is also true that a large number of QTLs are false positives or false negatives, environmentally dependent, and explain only a minor part of the variability observed. SNP and GWAM data are also problematic in many cases, and, similarly to QTLs, they typically explain only a smaller part of the phenotypic variability and likely have lower reliability. However, this problem has not been widely discussed in the existing literature. Most articles have concentrated on the visual symptoms, only recently coming to the realization that ear rot resistance and aflatoxin contamination are well-correlated in a number of genotypes, while, in others, the case is different. We consider all data to be highly reliable. It is uncertain whether current phenotyping methods are really sufficient to allow for solid genetic conclusions to be drawn from experimental data. Even so, most of the data refer to mainly additive inheritance, while, to a lesser degree, epistatic effects have also been reported. The big picture is possible in this light, but the conclusions for individual cases might be problematic. The existing data do not address the issue of immunity against *A. flavus*. However, the breeding of low-risk and stable hybrids is possible with the present knowledge. According to breeders, validated and effective genes or germplasms are required. To make such measurements, we have to prove their stability. For this, we need much more data than we have obtained to date. With increasing experience, the risk can be further decreased. At present, several resistance gene candidates show promise.

### 5.5. Common or Diverse Resistance Mechanisms against Ear Rot Fungi

The idea of the common resistance was implicitly included in the idea of resistance to natural ear rots. In wheat, such a common resistance has been proved [86,87,88]. For this reason, it is interesting to consider whether such a case is possible in maize. However, it is more important to determine how can we build up a screening system in which all three major pathogens are well-represented. This idea is not new. In our previous review article [8], several papers were cited which sought relationships between *F. graminearum* and *F. culmorum*, *F. graminearum* and *F. verticillioides*, and *F. verticillioides* and *A. flavus*. In every paper, low-to-moderate correlations were found, but no general consensus was reached. Except us nobody have dealt with the resistance to all three toxigenic species and their toxin relations. 

The only positive exception was the correlation series between resistance to *F. graminearum* and *F. culmorum*, which was consequently close and highly significant [10,48,52,73,184], indicating the similar resistance background with respect to these fungi. This is valid also for resistance to DON accumulation.

Bolduan et al. [17], (as shown in their Figure 3 [17]), observed a phenotypic correlation between *F. graminearum* and *F. verticillioides* symptoms (r = 0.63, *p* = 0.05). Looking at the plotted data, it is clear that, for up to 70% GER severity, no significant FUM contamination was present; meanwhile, inbreds having near 100% GER varied between 0 and 40% FER. For mycotoxins, the original data were not given (only the ln mg/kg data), and the correlation between DON and fumonisin contamination (ln mg/kg) reached r = 0.59 (*p* = 0.05). Of the hybrids, three were of types having very different responses for the toxins; for example, an inbred at DON 6.8 presented no fumonisin contamination. This indicates that a significant part of the inbreds behave similarly with respect to the two toxins, while the rest did not. Therefore, the behavior of DON cannot be forecast with the accuracy required for a breeding program. Other researchers [185] have identified a 2OGD superfamily gene group, which was detected as having similar functions against the two ear rot pathogens. This explains the similarities in resistance to the two fungal species in several genotypes, as also detected previously [50]; however, this result was not consistent for the toxin response or in another hybrid group with an entirely different genetic background. As the paper did not present aflatoxin data, the food safety aspects remain unclear. Cary et al. [4] suggested combining the resistance of African and USA lines with respect to *A. flavus*, *F. verticillioides*, and others. The beta-glucuronidase (*GUS*) reporter gene has an influence on *A. flavus* resistance in maize. The GER and FER resistance additive and dominant effects were predominant in most environments in response to these diseases and their toxins. However, the stability of their performance was low. Different genetic–environmental interactions were observed, indicating some degree of common resistance. Chiuraise et al. [23] have found, in crosses of fumonisin- and aflatoxin-resistant lines, that several lines had good resistance against both toxins. A dissimilarity between Hungarian and Kenyan data was the higher infection severity in response to both pathogens: fumonisin maxima were higher in Kenya, while aflatoxin maxima were much higher in Hungary [10,52]. In a larger number of hybrids, several were identified to have common resistance, but the majority of the hybrids responded differently [10,52]. A meta-analysis of QTLs and candidate genes was conducted in 224 QTL from 15 papers in a dense genome-wide SNP analysis and identified a total of 40 MQTLs (metaQTLs) [186], of which 29 were associated with 2–5 FER- and/or GER-related traits, 28 were common for FER and GER, and 19 were common for silk and kernel resistance. Promising candidate genes were also identified. As mycotoxin responses were not assessed, the value of the work in this respect remains unclear. The main message of this work is that QTLs with distinct functions exist, making the problem even more complicated but potentially explaining the varying responses to different *Fusarium* species and inoculation methods.

This is similar to the results that we obtained in wheat [135,136] and later in maize [51]. Guche et al. [25] identified the *ZmLOXs* genes participating in oxylipin biosynthesis as taking part in the resistance to *F. verticillioides* and *A. flavus*. *ZmLOX4* seems to be the most effective, which was identified in the inbreds Tzi18, Mo17, and W22. Oxylipin synthesis is connected to linoleic peroxidation by 9-LOXs and their accumulation. Triglyceride peroxidation was only observed in genotypes with resistance to *F. verticillioides*. The mechanisms were remarkably similar against the two fungal species, and they also were found to influence the mycotoxin content. Stagnati et al. [49] tested a hybrid collection against *F. verticillioides* and *A. flavus*. Regarding the symptoms, seven presented resistances to both pathogens, while, for toxins, only three were superior. These findings support the Hungarian results. Bolduan [17] obtained a moderate (r_p_ = 0.59, *p* = 0.05) (r_p_ = correlation, phenotypic) to moderately high (r_g_ = 0.77) correlation (r_g_ = correlation, genetic) between the two toxin reactions of the inbreds. Another paper [187] reported on crosses using the inbreds Co430 (*F. graminearum*) and Mp420 (*A. flavus*) with a susceptible inbred and backcrossed. Their results suggested that the best lines selected from these crosses also had resistance to *F. verticillioides*. The authors hoped that this result would indicate general significance. Abbas et al. [37] did not find any correlation between aflatoxin and fumonisin contamination; however, with other plant materials, significant positive or medium correlations were obtained [165]. This seemingly contradictory situation is the same in Hungary, where the results changed from test to test.

On the other hand, the selection of S5 inbreds resulted in several cases besides increased resistance to *F. verticillioides*, such as resistance to *F. graminearum* [188]. Guo et al. [26] screened 87 inbred lines mostly from China and Mexico and tested them for preharvest aflatoxin and fumonisin contamination. The aflatoxin level ranged from 50 to 1524 mg/kg, while the fumonisin level varied between 0.6 and 124 mg/kg (natural infection). Five lines showed very low aflatoxin content (artificial), four presented very low fumonisin content (ELISA test was used), and only one (TUN61) showed good resistance to both. The data support the conclusions we obtained from maize hybrids [10].

We reported a common resistance to *F. graminearum* and *F. culmorum* [189], which later tests further confirmed [50,52].

The resistances to *F. graminearum*, *F. verticillioides*, and *A. flavus* have been assessed over the last 10 years. We decided to test hybrids instead of inbred lines, as we aimed to help growers by determining which hybrid has high, low, or medium risk with respect to disease and toxins. The results clearly indicated that a common resistance does not exist to the three main pathogens in maize; therefore, resistance should be evaluated separately. In several tests, significant correlations were found between resistance responses while, in others, no correlation existed. The correlations between toxin contamination were sometimes exceptionally high, and, in other tests, no significant results were obtained. Until 2020, only two isolates had been tested together for a toxigenic species; since 2021, three isolates have been tested [10,50,52]. Through these tests, the stability of the responses could be determined. Despite the fact that the hybrids showed very variable resistance to disease and toxin accumulation, hybrids with low and stable resistance, even to all toxigenic species and their toxins, could be identified.

The differences in resistance tend to be highly significant, with the difference between minimum and maximum values often being higher than ten-fold. This makes it possible to identify more resistant hybrids, which should be preferred during variety registration, by screening commercial varieties. On the other hand, the exclusion of the highly susceptible hybrids will result in a sharp improvement of food and feed safety, and, through increasing breeding activities, even better control may be possible. This contributes to food and feed security, and significantly more grain can be used for food and feed and not lost.

### 5.6. Resistance to Other Toxigenic Fungi and Their Toxin Accumulation

As severe shortages in resistance testing and toxin risk evaluation regarding the major three ear rot pathogens, these issues should be resolved first. However, at the basic research level, work should be started to develop methods and gain new insights that will help us to control these toxic species and the levels of their toxins.

## 6. Influence of Climate Change on the Resistance Breeding Activity

Resistance to ear rot is complex, and the expression of resistance against disease and toxin contamination depends both on short- and long-term changes in climate. Higher aflatoxin contamination occurs in drier and hotter areas, while fumonisin prefers warm, humid conditions [9,190,191]. Furthermore, deoxynivalenol contamination generally increases in warm and humid seasons. It would be reasonable to think that the resistance to toxigenic fungi receives attentions when planning the reduction of mycotoxin contamination and its practices; however, this is not so. Zanon et al. [192] analyzed the control of mycotoxins. Many agrotechnical, biocontrol, and other practices were analyzed, but resistance to these fungi was not mentioned. Yu et al. [193] forecast significantly higher aflatoxin contamination for the Corn Belt for 2030–2040, where only sporadic contamination occurs at present. In the South, the situation mostly remains, and several counties are expected to have temperatures which are high enough to inactivate the fungi. Interestingly, the possible role of higher resistance in decreasing aflatoxin contamination has not been discussed. The case for Europe is similar [194]. The paper by Palumbo et al. [195] is an exception, in which it was mentioned that resistance adds no immunity. Kawasaki [196] stated that only agronomic adaptation is required: instead of double cropping, only one crop should be cultivated. Cui and Xiie [197] foresaw some chance for short-term adaptation, but the possibilities for long-term adaptations are not yet clear. Many articles have concentrated on food security problems, while food safety remains under-addressed.

It is clear that the higher temperatures and drought will increase the aflatoxin pressure under the current resistance structure, with mostly susceptible hybrids prevailing. We believe that higher resistance to *F. verticillioides* and *A. flavus* may balance the effects of increasing temperatures. However, for this, resistance data that can modify the present forecasting methods must be obtained, and hybrid resistance factors should be added to relevant models. For this, we also require higher resistance against drought and heat, better water retention, and optimized biological conditions in the soil; that is, a better adaptation ability is the key for the genetic and plant production parts.

## 7. Methodology Aspects

Inoculum production. Most papers have used only a single inoculum, mostly coming from single isolates (Table 2). As conidial suspensions are typically used, the conidium concentration differs in different tests; however, an aggressiveness control is generally not used. Therefore, the conidium concentration lacks scientific content when compared to the concentrations applied in other articles. The lack of aggressiveness testing in the case when suspensions are used means that the original role of the regulation of the conidium concentration is forgotten, as the aggressiveness level is not regulated in the test. Many papers have reported on differences in aggressiveness within pathogenic populations. Notably, a given conidium concentration can present highly differing aggressiveness differences. A key problem is that, at low aggressiveness, the differentiation of genotypes is poor, and the data have very limited genetic value. Decades ago [198], we assessed the ear rot resistance of ten hybrids with respect to 14 *F. graminearum isolates*, and the plotted results (Figure 1) [51] clearly indicated the different responses of the hybrids to the isolates.

It is possible that all of the responses are true, and we think that this is correct. The fungal population in a field consists of millions of spores, conidia, or ascospores. Each isolate differs, more or less, from the others—in this case, in terms of aggressiveness. The differences between hybrids are dependent on aggressiveness. At high aggressiveness, differentiation is much better. When more isolates are used, their mean response is closer to reality. Of course, this requires more work. The correlations between hybrid responses to the 14 isolates (*n* = 12) show that, out of the 45 correlations, 22 were not significant, 18 were significant at *p* = 5%, 4 were significant at *p* = 1%, and 1 provided significance at *p* = 0.1%. From the correlations (*n* = 91), 65 were not significant, 14 were significant at *p* = 0.05, 10 were significant at *p* = 1%, and 2 had significance at *p* = 0.001. Our later data show similar results [10,50,52]. This indicates a problem in resistance breeding, genetic studies, and variety registration. This is the reason why—in addition to the many other similar experiences—we do not trust in the one-inoculum tests. As the topic seems to be important, more research is needed.

It seems that the ln function/ln = log_e_/transformation [17] should be used very carefully. By neutralizing some effects, we may introduce unfamiliar problems into the analysis. In the author’s opinion, it is not clear whether the genetic background could be described any better in this way. Therefore, three independent isolates are preferable when testing a mapping population, as well as conducting a QTL analysis for each isolate independently and checking the mean results of the isolates.

Resistance to ear rot means evidencing a significant difference in symptom severity between genotypes across years, locations, and/or isolates. A key task is to determine which symptom is to be measured or evaluated. Breeders have preferred for decades to conduct assessments based on natural infection; however, it has slowly become clear that consequent breeding work is hardly possible [8]. Artificial inoculation methods have existed for many decades, with the toothpick method going back to Young [199], who first described this methodology, which has been modified many times since. Later, the silk channel method became widespread [200,201], following an article that stressed the role of silks in ear rot development as a primary source [120,121]. Both approaches have many subversions [8]. We used toothpicks inserted into holes made using metal nails (15 mm long and 1.5 mm diameter); for further details, see [50,52,198]. The inoculation is typically made in the middle of the ears after mid-silking at 6–15 days; however, this is variable [200,201]. Mid-ear inoculation can also be carried out by injecting a *Fusarium* or *Aspergillus* suspension into a hole on the ears or by using a steel needle submerged into a pathogen suspension before inoculation. Munkvold et al. [202] concluded that the silk channel inoculation is closer to the main infection pathway; therefore, this is now the most commonly used inoculation method. The response of the plant is called silk channel resistance. Presello et al. [203] compared the kernel and silk channel methods and found that kernel inoculation resulted in significantly higher infection severity than the silk channel method, and the kernel infection data were significantly more stable. The use of inoculum to spray the silks has also been suggested, termed as silk resistance [204]; however, this was significantly less effective in our praxis, and even an ensured high humidity after silk inoculation using polyethylene bags did not help [8]. The correlations indicated between kernel and silk resistance a more common background behind them as normally supposed, with r = 0.60 [52]. It is also clear that a resistance program cannot use two different methods for resistance screening at the same time. The kernel test severity and toxin contamination were more than three-fold higher under the dry and warm Hungarian conditions than the silk channel inoculation results. This was true for all three *Fusarium* spp. tested (*F. graminearum*, *F. verticillioides*, and *F. culmorum*). Possible explanations are the cobs overgrowing the husk leaves and the early opening of husk leaves helping to dry out the grains, leading to decreased infection. As a 1–5 mL suspension is injected in the silk channel method, different infection areas are possible in the ears; therefore, the symptoms are generally more variable and less reliable [204]. For the highly aggressive *F. graminearum*, this might be less important; however, for the significantly less pathogenic *F. verticillioides* and *A. flavus*, this may be a more significant factor.

The use of kernel inoculation methods also resulted in a much higher *Aspergillus* infection severity and aflatoxin contamination, and several were unacceptable [181]. It was concluded, at that time, that no inoculation method can be developed that remains stable across environments, as, in their tests, no stable resistant or susceptible inbreds or hybrids were identified, and a comparison of the testing methods proved to be problematic. In another test [37], the kernel inoculation method was preferred over the silk-related part of the infection process in order to set better controls for post-infection differences in the hybrids. In the author’s opinion, the inoculation methods cannot be blamed for the inconsistency of the received data. In these sources, there was no mentioning of the difference in aggressiveness in the resistance testing or the relationship between aggressiveness and toxin production on the one hand and the relationship between resistance to toxin accumulation on the other hand. Interestingly, medium-ripening (flexible) hybrids showed the largest adaptation ability, but not without exception. Poulsen et al. [205] compared five ear inoculation methods and came to the same conclusion. Under dry and warm conditions, the symptom severity and toxin contamination were 5–10-fold higher than when using silk channel inoculation [52]. Sampling of the inoculated material can cause severe problems [10,51]. It was not mentioned that the isolates may have a varying influence on the disease process and toxin contamination [10]. Furthermore, the ecological and environmental conditions influence the fate of the infection [191]. The author agrees with the conclusion that a better understanding of the disease development and epidemic behavior is required.

The evaluation of symptoms. Reid et al. [44,201] assessed GER and FER on a scale ranging from 1 to 7. Cao et al. [19] used a scale from 1 to 9, where 9 indicated no damage and 1 indicated infection severity higher than 90%. Here, the healthy part of the ear was rated. Mesterhazy [19], however, used a percentage scale between 0 and 100% for ear rot evaluation, which has been significantly refined in recent years, after toxin contamination was considered very important for the comparison with ear rot severity [10,51,52,126]. For ear rot evaluation, this was feasible for *F. graminearum*, for which the highest severity values mostly varied between 0 and 100%; however, when the maximum was only 5–10% for a less aggressive isolate or dryer year, every genotype appeared to be resistant, and the distribution was between 1 and 3 according to the scale of Reid et al. [44]. As *F. verticillioides* is much less pathogenic (10–15% of the hybrid mean for *F. graminearum*, while *A. flavus* again has 10–15% of the mean for *F. verticillioides*), the correct evaluation of lower infection values becomes very important. As the means for measuring mycotoxin contamination requires a careful evaluation using the same scale in tests where all three resistances are tested at the same time, a fine percentual evaluation (differentiation between 0.1 and 1% also) was further introduced for all pathogens and experimental types [10,51].

Sampling is another key problem; a much better sampling method was developed [51,52] in which five selected ears are shelled and the whole grain yield (about 1 kg) is roughly milled. In this way, several grains that contain aflatoxin, or any other toxin, are broken into 1 mm (or somewhat larger) pieces which can be mixed. In this way, any sampling errors can be reduced significantly. From the obtained material, 100 g is separated for further analysis, and from this, a 10 g amount will be finely milled. There are many arguments for the multi-toxin analyses also including the masked toxins or their variants, like the acetylated versions of DON and the a and b versions of ZEN [206,207]. From these sampling procedures, we can obtain more reliable toxin results.

## 8. Resistance and Agronomic Practices

An increased resistance level can be considered to be highly important. The reason for this is that, in a production chain, the individual elements combined do not interact independently, as a spider in the middle of a web collects all information and acts accordingly. We have seen that wounds significantly increase toxin contamination by fumonisins and aflatoxin, while *Bt* genes support hybrids with lower toxin accumulation. However, susceptible hybrids can have very high disease severity and toxin contamination without any insect-mediated wounds. Therefore, a combination of genetic modification (GM) and resistance to disease and toxin accumulation can be expected to result in better protection against fumonisins and aflatoxins. It often occurs that, in papers focused on managing aflatoxin contamination, resistance to *A. flavus* and related fungi is not mentioned. We do not have effective fungicides and fungicide technology for highly susceptible hybrids; the experimental proof of this is that there has been no success. However, protecting the more resistant ones might bring better results, as has been achieved in wheat [53,54,208,209]. These tests are yet to be conducted, but they might bring about better protection, although only in the moderately or highly resistant hybrids. The use of more resistant hybrids and varieties may increase the efficacy of the application of nontoxic *A. flavus* strains (like Aflasafe) on more resistant plants, for which the combination is feasible. Many papers have reported on the aflatoxin-increasing effects of drought and heat. In this context, the use of soil-loosening technology down to 60 cm (if possible) with biologically active soil organisms that can supply nutrients can provide significant help when good or highly resistant varieties and hybrids are grown; furthermore, drought- and heat shock-resistance breeding may also be required. Disease and toxin forecasting also requires resistance data with respect to diseases and toxin accumulation in order to better forecast the toxin risk, especially when hybrids with good resistance reach a significant market position. This is also important for plant protection, as more-resistant plants will be less receptive to infected plant debris.

Overall, we need more resistant plants and a variety registration network that excludes susceptible and highly susceptible hybrids or varieties; furthermore, we need a much better adaptation ability of plants to facilitate disease and abiotic resistance in order foster much more successful maize production. This is the opposite case as seen at present, as higher grain yield ability alone cannot save production. A better-adapted and resistant material can reach significantly higher and healthier grain yields than any varieties with one-dimensional high productivity, which may lead only to increased losses.

## 9. Breeding Aspects

Although this refers to preharvest contamination, the mixed scenario (i.e., pre- and postharvest data) presents a similar picture [210]. Based on the multiple infection and multi-toxin data from the field, foods and feeds, blood, and urine samples, the food safety task is much more severe challenge than what we thought before, think now, and will probably think in the future. We need complex resistance to them and their toxins as far as possible, and the molecular genetics can provide better solutions [210,211]. As the seasons have a tendency to become warmer [212,213,214], the higher resistance to *A. flavus* and *F. verticillioides* will have increasing significance. The ecology will prefer higher toxin contamination. Without preharvest control (better adaptation and better agronomy), we cannot have too much hope. However, this can be balanced, and the whole production system has to be reformed.

The elements of developing an improved methodology for building resistance to ear rot and toxin accumulation were discussed in detail previously [10,51]. As the variety rank and mean infection severity vary between isolates, the use of at least two (or, even better, three) isolates without mixing is suggested. As resistance does not have the same genetic background with respect to different forms of ear rot, resistance must be tested for each pathogen separately. As the toxin response has other genetic regulatory elements beyond resistance level, a toxin analysis must be conducted in all relevant scientific and applied research; otherwise, the toxin contamination risk cannot be regulated. Natural infection provides important feedback for breeders and growers, as their efforts will be accepted when the harvested grain yield corresponds to the forecasts. Therefore, at harvest, every truck should be tested for toxin contamination, providing the possibility to store the differentially contaminated lots separately. Multi-toxin approaches of checking for natural infection and toxin contamination should be introduced as soon as possible. At present, there exist methods which require only a few minutes to perform such work.

A better methodology for genetic work is also required in order to provide more reliable results. With an improved methodology, the LOD values could be increased by 2–3-fold, compared with traditional methodologies in wheat. The basic tests should also be conducted for maize. At present, modern genetic analyses produce a lot of important information; however, they also require a methodical background to improve phenotyping and provide reliable information on the resistance level associated with given genes. For breeding, the combining ability of the inbreds is decisive for grain yield and ER resistances.

A better methodology is very useful for growers, as they can receive data that include all known and unknown resistance factors, that is, the total resistance of the plant. It is suitable to provide better data for new genes in basic research, gene combinations, and so on, in the hope that, by decreasing the background noise, the same genes or QTLs will receive larger LOD values, and the explained variance can be doubled, as has been performed in wheat [135,136], and the number of detected QTLs may grow by 2–3-fold.

At present, even though the provided information is rapidly growing, most breeding firms do not provide reliable information on the resistance characteristics of their hybrids. As permission has been received to introduce a new variety registration system in Hungary, financing remains the only problem. According to calculations, the variety registration cost is about EUR 2.5 million for all small grain and maize variety candidates with toxin analyses; on the other hand, the revenue may surpass the EUR 100 million. From an economic viewpoint, this is not a bad business.

## 10. Conclusions

In spite of the progress made in research, a decrease in mycotoxin contamination has not taken place in the fields. While the discussion on the necessity of toxin tests has been ongoing for a long time, new facts have slowly provided an answer: without toxin data, an assessment of food safety is not possible. Over time, significant differences between hybrids have been reported with regard to the determination of key toxigenic species, indicating that the screening of the hybrids is a reasonable activity, as 10-fold (or even higher) differences in resistance have been determined between registered commercial hybrids. For the risk analysis of hybrids, artificial inoculation and toxin data are required for the three main pathogens, as well as natural infection and corresponding data (if possible, the most important six targeted toxins could also be screened for the detection of unknown risk factors), in comparison to non-inoculated control rows. We identified highly resistant and highly susceptible hybrids from the same breeding firms, but their genetic background is unknown. This clearly demonstrates that low toxin contamination is rather an accident and not the result of consequent breeding work. By conducting serial screening, firms could identify superior genotypes in a relatively short time, especially when state registration will make this qualification necessary. Furthermore, toxin analyses should be introduced into genetic research, as higher resistance does not always ensure lower toxin contamination. Key candidate genes require validation through use in breeding programs. Most of the QTLs identified thus far have a small effect, but several candidate genes seem to be more effective; however, their validation will take time. There is a need to select for low symptom severity and low toxin contamination; about one-third of the tested hybrids appeared to belong to this group, while a smaller part (10–15%) presented low risk for all fungi and relevant toxins tested.

## Figures and Tables

**Figure 1 jof-10-00040-f001:**
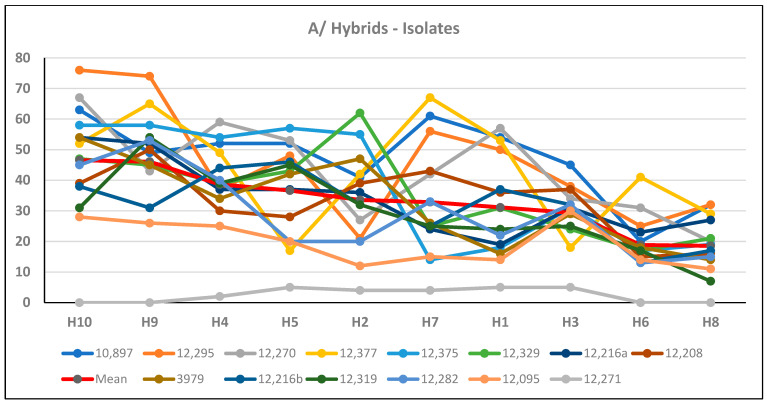
The GER kernel infection (toothpick method) severity of 14 *F. graminearum* and *F. culmorum* isolates in maize hybrid resistance tests with 10 hybrids (H1–H10) [10]. The read mean line shows the mean response of the hybrids and isolates. (**A**). It is considerable that H10 and H9 have the same mean infection severity; their responses to different isolates strongly vary as for isolates 10,897 and 12.295. We have the same deviation for H2 and H3, and H9 and H7 for isolates 12216b and 12,319. (**B**). We see the same variance in the responses of the hybrids. The means of 10,897 and 12,295 are nearly the same, but the infection severity of 12,295 is significantly higher than that found for 10,897. In the case of H2, we have a response similar to the mean for 10,897, but a rather low infection severity to 12,295. The differences are not necessarily coming from race specificity; nobody proved this until now. Comparing the two inocula of 12,216, we see their different behavior. This is the case also in wheat [86].

**Table 1 jof-10-00040-t001:** Methodical and evaluation data of publications cited for maize *F. graminearum*, *F. verticillioides*, and *A. flavus* resistance tests.

Authors	Ref. No.	Pathogen	Inoculation	No. of Isolates/mixture	Conidium conc./mL	Aggressiveness	Evaluation	Severity% min–max	DON mg/kg	FUM mg/kg	AFB1 mg/kg
Afolabi et al., 2007	[14]	F. vert.	Art. silk	1	1 × 10^6^	No		2.3–57		1–53	
Araujo et al., 2022	[15]	F. vert.	Silk + kernel	1	5 × 10^5^	No	Scale 1–9	2.5–6.5		No	
Bernardi et al., 2018	[16]	F. vert	Art. kernel	1	1 × 10^6^	No	Not tested	No		No	
Bolduan et al., 2009	[17]	F. vert.	Art. silk	1	1 × 10^6^	No	%	1.6–39.5		0.05–58	
Butron et al., 2019	[18]	F. vert	Art. silk	1	1 × 10^6^	No	Scale 1–7			No	
Butron et al., 2019	[18]	F. vert	Art. kernel	1	No data	No data	Scale 1–7	No		No	
Cao et al., 2014	[19]	F. vert.	Natural	-	-	-	% disinfect.	4.6–35.3		No	
Cao et al., 2014	[19]	F. vert.	Natural	-	-	-	%, no disinfect.	67–84		No	
Cao et al., 2014	[20]	F. vert	Art. silk	1	2.5 × 10^5^	No	Scale 1–9	No		4.1–5.9	
Cao et al., 2014	[20]	F. vert	Art. kernel	1	2.5 × 10^6^	No	Scale 1–9	No		2.6–9.6	
Cao et al., 2022	[21]	F. vert.	Art. kernel	1	1 × 10^6^	No	Scale 1–8	No		No	
Chen et al., 2016	[22]	F. vert.	Art.	1	5 × 10^6^	No	Scale 1–7	0–74		No	
Chiuraise et al., 2016	[23]	F. vert.	No data	No data	No data	No data	Scale 1–7	2–25		22–48	
Gesterio et al., 2021	[24]	F. vert	Not given	No	1 × 10^6^	No		No		0.02–0.2	
Guche et al., 2022	[25]	F. vert	Art.	1	1 × 10^6^	No		3.5–50		2.37–179 ppb	
Guo et al., 2016	[26]	F. vert.	Natural	No data	Not appl.	No				0.3–5	
Henry et al., 2009	[27]	F. vert	Art. kernel	1	9 × 10^7^	No	cm^2^	1–9		2–48	
Ivic et al., 2008	[28]	F. vert.	Art. kernel	7	3 × 10^6^	No	Scale 1–7	2.29–5.76		No	
Lanubile et al., 2014	[29]	F. vert.	Art.	1	3.5 × 10^6^	No	No data			No	
Lanubile et al., 2014	[30]	F. vert.	Art. kernel	1	3.5 × 10^6^	Tested	No			No	
Lanzanova	[31]	F. vert.		2	1 × 10^6^	No	No inf. Grains	4.7–97/cob		4.4–141	
Loeffler et al., 2010	[32]	F. vert.	Art. silk	1	1 ×10^5^	No	%			No	
Santiago et al., 2013	[33]	F. vert	Art. kernel	1, 2 mL	1 × 10^6^	Old data	Scale 1–7	1.3–6.04		0.1–1.5	
Schjoth et al.	[34]	F. vert.	Art. silk	6	1 × 10^6^	FB content	Scale 1–7	No		60–1083 ppb	
Wu et al., 2020	[35]	F. vert	Art. silk	1	1 × 10^6^	No	Scale 1–7			No	
Zila et al., 2014	[36]	F. vert	Art. silk	Not given	2 × 10^6^	No				No	
Abbas et al., 2012	[37]	*A. flavus*	Art. kernel		1 × 10^6^	No	No data	No			1–3008
Abbas et al., 2012	[37]	*A. flavus*	Natural	Not appl.	Not appl.	Not appl.	No data	No			0–289
Chiuraise et al., 2016	[23]	*A. flavus*	No data	No data	No data	No	Scale 1–7	1.5–40			1.9–32
Guche et al., 2022	[25]	*A. flavus*	Art.	1	2 × 10^6^	No		5–71			12–30 × 10^3^
Guo et al., 2016	[26]	*A. flavus*	Silk	1	4 × 10^6^	No					52–1524
Henry et al., 2009	[27]	*A. flavus*	Art. kernel	1	3 × 10^8^	No	cm^2^	1–9			18–10,592, mg/kg *
Henry et al., 2012	[38]	*A. flavus*	Art. kernel	1	3 × 10^6^	No	0–9	0.3–7.3			0–776
Williams et al., 2014	[39]	*A. flavus*	Art. kernel	1	3 × 10^8^	No	No	No			0–119
Ali et al. 2005	[40]	F. gram	Art. kernel	No	2.5 × 10^6^	No	Scale 1–7	No	No		
Ali et al., 2005	[40]	F. gram.	Art. silk	No	2.5 × 10^6^	No	Scale 1–7	No	No		
Bolduan et al., 2009	[17]	F. gram	Art. silk	1, IFA66	1 ×10^5^	No	%	22–100	0–3072		
Gaikpa et al., 2021	[41]	F. gram.	Art. silk	1	1.5 × 10^4^	No	%	0–90	No		
Galiano-Carnerio et al., 2021	[42]	F. gram	Art. silk Br	3	5 × 10^4^	No		n.d.	No		
Galiano-Carnerio et al., 2021	[42]	F. gram	Art. silk De	1	1.5 × 10^4^	No	%	0–68	No		
Kebede et al., 2016	[43]	F. gram	Art. silk	2	No	No	%	8–92	No		
Kebede et al., 2016	[43]	F. gram	Art. kernel	2	No	No	%	12–82	No		
Loeffler et al., 2010	[32]	F. gram	Art. silk	1	No	No	%	28–83	49–1009		
Reid et al., 1996	[44]	F. gram.	Art. silk	No	Unknown	No	Scale 1–7	No	Yes		
Reid, Hamilton 1996	[45]	F. gram	Art. kernel	No	0–2 × 10^6^	No	Scale 1–7	No	No		
Zhou et al., 2021	[46]	F. gram.	Art. kernel	1	1 × 10^6^	No	Scale 1–7	2.2–8	No		
Schaafsma et al., 1997	[47]	F. gram.	Silk	1	1 × 10^5^	No	Scale 1–7	S–MR	1.4–4		
Schaafsma et al., 1997	[47]	F. gram	PIN	1	1 × 10^5^	No	Scale 1–7	S–MR	2–4.6		
Schaafsma et al., 1993	[48]	Fg, Fv, Fm	Silk	2 + 2 + 21	No data	No	Scale 1–7	1.7–4.6	1–15.4	0–11.5	

* mg/kg in the original paper, possibly μg/kg, F. gram, *F. graminearum*; F. vert, *F. verticillioides*; art., artificial inoculation; silk, silk channel method; kernel, kernel resistance.

**Table 2 jof-10-00040-t002:** Global maize production [56] and its losses counted from the data of Mesterhazy et al. [57].

Item	Yield MMT	Y * Total %	Y ** Harv. %
Total capacity	1590	100	
Total harvested	1060	66.7	100
Preharvest loss (biotic, abiotic, 3% for harvest) 33%	530	33.3	-
Mycotoxin contamination at harvest	106	6.7	10
Storage waste	212	13.3	20
Consumer and other waste	138	8.7	13
Total waste	986	62.0	43
Total grain used	604	38.0	57

* Y = grain yield. ** Grain yield harvested.

**Table 3 jof-10-00040-t003:** Mycotoxigenic *Fusarium* species associated with cereals and their mycotoxins [73].

Fusarium spp.	Mycotoxins ^a^
*F. acuminatum*	T2, MON, HT2, DAS, MAS, NEO, BEA
*F. anthophilum*	BEA
*F. avenaceum*	MON, BEA
*F. cerealis*	NIV, FUS, ZEN, ZOH
*F. chlamydosporum*	MON
*F. culmorum*	DON, ZEN, NIV, FUS, ZOH, AcDONs
*F. equiseti*	ZEN, ZOH, MAS, DAS, NIV, DAcNIV, FUS, FUC, BEA
*F. graminearum*	DON, ZEN, NIV, FUS, AcDONs, DAcDON, DAcNIV
*F. heterosporum*	ZEN, ZOH
*F. nygamai*	BEA, FB1, FB2
*F. oxysporum*	MON, BEA
*F. poae*	DAS, NIV, FUS, MAS, T2, HT2, NEO, BEA
*F. proliferatum*	FB1, BEA, MON, FUP, FB2,
*F. sambucinum*	DAS, T2, NEO, ZEN, MAS, BEA
*F. semitectum*	ZEN, BEA
*F. sporotrichioides*	T2, HT2, NEO, MAS, DAS
*F. subglutinans*	BEA, MON, FUP
*F. tricinctum*	MON, BEA
*F. verticillioides*	FB1, FB2, FB3

**^a^ Abbreviations:** AcDONs—mono-acetyldeoxynivalenols (3-AcDON and 15-AcDON); AcNIV—monoacetylnivalenol (15-AcNIV); BEA—beauvericin; DiAcDON—di-acetyldeoxynivalenol (3,15-AcDON); DAcNIV—diacetylnivalenol (4,15-AcNIV); DAS—diacetoxyscirpenol; DON—deoxynivalenol (vomitoxin); FB1—fumonisin B1; FB2—fumonisin B2; FB3—fumonisin B3; FUP—fusaproliferin; FUS—fusarenone-X (4-acetyl-NIV); FUC—fusarochromanone; HT2—HT-2 toxin; MAS—monoacetoxyscirpenol; MON—moniliformin; NEO—Neosolaniol; NIV—nivalenol; T2—T-2 toxin; ZEN—zearalenone; ZOH—zearalenols (α and β isomers).

**Table 4 jof-10-00040-t004:** Toxin production for a percentage of artificial inoculation tests in 2019/2020 (based on Table 6 of [10]).

Hybrid	Rates between Toxin (mg/kg)/Ear Rot%			Mean	Variance
	DON/GER%	FB1/FER%	AFB1/AER%		
**Sy Talisman ***	** 0.554	4.500	0.151	1.74	5.77
**Armagnac**	1.277	3.783	0.367	1.81	3.13
**SY Zephir**	0.837	4.730	0.263	1.94	5.91
**Konfites**	0.605	5.612	0.413	2.21	8.69
DKC 4541	1.983	2.484	2.967	2.48	0.24
**Konfites**	1.788	5.721	0.154	2.55	8.19
Illango	2.345	4.943	0.622	2.64	4.73
Kleopatras	2.119	4.114	2.552	2.93	1.10
P9415	1.534	2.300	5.573	3.14	4.60
P9718E	4.621	3.102	3.100	3.61	0.77
Koregraf	1.848	5.881	3.733	3.82	4.07
P0725	1.796	9.156	0.557	3.84	21.61
ES Lagoon	1.967	9.719	0.891	4.19	23.20
DKC 5830	2.753	6.250	4.100	4.37	3.11
ES Harmonium	3.272	9.130	0.842	4.41	18.15
Sy Zoan	2.725	14.170	0.322	5.74	54.76
Valkür	2.871	15.500	0.136	6.17	67.17
Korimbos	5.302	40.500	7.863	17.89	385.11
**Mean**	**2.23**	**8.42**	**1.92**	**4.19**	**34.46**
Max/Min rate	8.69	17.61	57.93		

* Bold: Low and stable rate for the three toxins. Only dark and low green classification. ** Dark green, lower than 50% of the mean, low risk; light green, 51–100% of the mean, low-to-medium risk; yellow, 101–150% above mean high risk; orange, above 150%, very high risk.

**Table 5 jof-10-00040-t005:** Summary of the maize hybrid resistance test to GER, FER and AER pathogens and their toxins, artificial inoculation data 2019/2020 across two isolates per pathogens [10].

Hybrid		Ear Rot Severity			Toxin Content		ER Severity		Toxin Content Check			
	Fg %	Fv %	Afl %	DON mg/kg	FUM B1+B2 mg/kg	Afla ppb	F. Check	Asp. Check	DON mg/kg	FUM B1+B2 mg/kg	Afla ppb	FAO No.
Konfites	10.15	0.61	0.35	18.15	3.49	53	0.21	0.000	1.70	1.88	2	430
ES Harmonium	11.49	0.23	0.15	37.60	2.10	62	0.25	0.000	0.13	0.81	4	380
Sy Talisman	13.50	0.40	0.27	7.48	1.80	71	0.27	0.000	0.72	0.54	2	250
Korimbos	14.35	0.14	0.12	76.08	5.67	44	0.08	0.000	3.49	0.30	408	575
P0725	15.91	0.45	0.11	28.58	4.12	613	0.17	0.005	0.00	0.12	794	560
Koregraf	16.62	0.42	0.24	30.72	2.47	37	0.10	0.000	0.00	0.30	352	410
DKC 5830	17.08	0.52	0.44	47.02	3.25	245	0.17	0.000	2.18	5.63	0	560
ES Lagoon	19.76	0.32	0.24	38.87	3.11	896	0.19	0.005	0.00	2.23	20	460
Armagnac	19.79	0.23	0.11	25.27	0.87	98	0.13	0.005	2.33	1.58	9	490
Illango	19.94	0.53	0.36	46.75	2.62	1068	0.18	0.008	0.05	0.75	1143	530
P9718E	21.86	0.49	0.25	101.01	1.52	638	0.09	0.000	0.17	1.17	3	390
Kathedralis	22.10	0.67	0.58	13.38	3.76	361	0.21	0.000	0.27	1.73	0	490
DKC 4541	24.59	1.61	0.54	48.77	4.00	174	0.30	0.005	3.59	3.06	0	370
Valkür	28.78	0.10	0.05	82.64	1.55	205	0.05	0.000	1.28	0.81	0	731
P9415	32.04	0.60	0.28	49.14	1.38	38	0.24	0.000	0.58	2.32	0	350
SY Zephir	33.92	0.37	0.19	28.38	1.75	160	0.16	0.005	0.63	2.36	4	390
Kleopatras	35.02	0.44	0.10	74.22	1.81	310	0.16	0.000	0.35	2.31	114	630
Sy Zoan	35.36	0.47	0.16	96.35	6.66	1258	0.18	0.008	0.00	1.65	0	560
Mean	21.79	0.48	0.25	47.24	2.88	352	0.17	0.002	0.97	1.64	159	
LSD 5%	8.60	0.55	0.15				0.08					

Blue cells highlight genotypes with lower data than mean. Fg, *F. graminearum*; Fv, *F. verticillioides*; Afl, *A. flavus*; Afla, aflatoxin B1; F., Fusarium; Asp., Aspergillus; FAO No., ripening group classification of maize hybrids according to Food and Agriculture Organization of United Nation’s Organization. Cell colors: dark green, lower than 50% of the mean, low risk; light green, 51–100% of the mean, low-to-medium risk; yellow, 101–150% above mean, high risk; orange, above 150%, very high risk.

## Data Availability

The data that support the findings of this study are available in the cited literature and from the corresponding author, upon reasonable request.

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
