# Peer review of "Food Safety Aspects of Breeding Maize to Multi-Resistance against the Major (Fusarium graminearum, F. verticillioides, Aspergillus flavus) and Minor Toxigenic Fungi (Fusarium spp.) as Well as to Toxin Accumulation, Trends, and Solutions—A Review"

_jof, 2024, doi:10.3390/jof10010040_

Round 1
Reviewer 1 Report
Comments and Suggestions for Authors
This paper summarized the effects of different maize on the resistance and toxicity of different fungi. The article mainly lists Methodical and evaluation data of some fungi, the ability of different fungi to produce toxins and different maize resistance tests to pathogens and their toxins. However, the innovation of this paper is insufficient. Therefore, more publications to be supplied for the conclusions and the result descriptions should be majorly modified for accurate implications. See the following details:
1. Simplify the abstract to make it more logical.
2. Please note punctuation errors,such as line45,line46 and Table 2 .
3. More data are needed to support the conclusion in line 188-189.
4. Please reformat line 1087-1088.
5. Please present the data of Figure 1 more intuitively.
6. The first two sentences of the conclusion are ambiguous.
7. The novelty of this article is not sufficient.
Comments on the Quality of English LanguageModerate editing of English language required
Author Response
Author's Reply to the Review Report (Reviewer 1)
Please provide a point-by-point response to the reviewer’s comments and either enter it in the box below or upload it as a Word/PDF file. Please write down "Please see the attachment." in the box if you only upload an attachment. An example can be found here.
* Author's Notes to Reviewer
Dear Reviewer 1,
Thank you very much for your very valuable work evaluating this review. Thank you for your good opinion and suggestions, where the paper can be improved. I have rewritten the whole abstract; I hope it is mor logical and consequent then it was before. My comments are printed with red to better differentiate from your text and I hope they will adequate responses to your comments and suggestions.
I am indebted for your suggestions and critic; they surely helped to make the paper better.
Yours very sincerely
Akos Mesterhazy
Comments and Suggestions for Authors
This paper summarized the effects of different maize on the resistance and toxicity of different fungi. The article mainly lists Methodical and evaluation data of some fungi, the ability of different fungi to produce toxins and different maize resistance tests to pathogens and their toxins. However, the innovation of this paper is insufficient. Therefore, more publications to be supplied for the conclusions and the result descriptions should be majorly modified for accurate implications. See the following details:
- Simplify the abstract to make it more logical.
The Abstract was rewritten. I hope, it is better now.
- Please note punctuation errors, such as line45,line46 and Table 2 .
They were improved. Thanks
- More data are needed to support the conclusion in line 188-189.
I added more data and more data to see more examples. For this review seems to be enough, otherwise a separate publication from the existing data since 2007 could be provided to analyze 15 years together. This is a task that should be done. OI checked several years from the 15, and with major or larger differences the picture is the same. The best yielding location gave normally twice or higher yield than the national average is. Even the worst performing hybrids are better 70-80% higher than the national mean is. Therefore, we say that the not the genetic yielding ability is responsible for the low yields, but inadequate natural condition, bad agronomy, and many other things. and of course, genetics, where the adaptation ability is very poor of hybrids with otherwise very high yielding ability. This knowledge is not new. When I was a student long ago, we were said the yield depends about one third on genetics, another third on natural conditions, climate, etc. and third again on agronomy, tillage, fertilization, plant protection. In spite of this I hear since decades the yield is the most important and nothing else. The only problem is that even the worst yielding variety can give 50-80% more yield than the national yield. The data have another lesson. Under good conditions when national yield is high like in Hungary in 2020, 2021 the difference between yield capacity is smaller compared to years where significant epidemics, draught or other problems reduced yield significantly. I mention only here the year 2022, when a very severe draught The nation mean was 3.47 t/ha, but the best location with irrigation gave a mean of 16.01 t/ha with a maximum 17.54 t/ha. However, for aflatoxin contamination was the most severe years we measure it. Beside this we realized very large differences in draught resistance. For example, in plots of 30 plants we had hybrid where only one ear was harvested with agreeable seed set , but other hybrid, the best gave 12-16 close normal ears. This shows surely also genetic differences and the lack or presence of adaptation ability of the hybrids. As this material is now in preparation for publication, I do not want to cite here the data, in a review we should keep us to the published materials. However, for information it might be important for you and also explants why the general adaptation of the hybrids has such a huge significance.
- Please reformat line 1087-1088.
Thanks, you are right. The sentence was not finished, and the cited sentence of Warburton was not positioned between quotation marks. These were added. The sentence was finished. I think, it will be OK now. These improvements are now in lines 1203-1204 of the improved version. It happens sometimes the line number make a spring between pages. Earlier I faced several times with this problem, my solutions were that I gave the new position of the improved text that the reviewer could identify easily the right location.
- Please present the data of Figure 1 more intuitively.
I agree. I changed the diagram with the same data and added another diagram with the difference that not the isolates, but the hybrids were in the X axis. I extended the description of the figures, explaining what the essence is we see here. but isolates are on the x axis. I give the explanation in the main text. I hope, this is clear now, why the isolate problem inhibits a clear vie in the resistance evaluation, and this influences all results where only one inoculum is used.
- The first two sentences of the conclusion are ambiguous.
They are corrected.
- The novelty of this .article is not sufficient.
Thinking on your remark, what is novelty in this paper. We are used to name only positive thing. In the fully revised abstract, I listed several points in resistance matters we face and make progress slower as it could be. The statement that the decrease in the preharvest toxin reduction on the field is not the case or an increase for example in aflatoxin in Hungary happens, can also be a novelty that raises the need to make or continue the research in increasing resistance. It seems me a novelty that the multitoxin data will give us new tasks as we should consider more toxins to control as we thought until now. I think, identification of problems to need new solutions can be a novelty. When I was a young fellow, I said once my chief who was an excellent scientist, that I wanted results and less interested in methodical matters. He said. Consider this, as 80 % of the Nobel prizes were given for methodical achievements. I think, he was right. In a review article only published results can be considered, on the other side it provider a possibility to sum up the work that has been done, and show possibilities for better solutions we have now.
But even for this study I screened more than 500 papers, and the selected ones contain smaller or larger positive or negative signs we collected here. I think, both are valuable. We also need proofs that the research leads to a blind alley. When this is sure, we leave the line. We have also positive results that bring us closer to an improvement. It is an important novelty that the genetic background to the three main toxigenic fungus is mostly different, therefore, the resistance to them separately should be tested. It can also be a novelty that QTLs were identified that functioned against different toxigenic species, some others were specific. The means that the experimental facts slowly can be explained, and this knowledge could be useful for selecting even better inbreds and hybrids in the future. It can be a novelty that the symptom and toxin correlations are not so close that the toxin contamination could be forecasted. When the EFSA sets toxin limits, nobody will be interested in the symptom severity, they want to have a toxin report below the limit. It seems to be a simple thing. In spite of the, most papers deal only with the symptoms. It is novelty that we realized that without toxin data a food safety declaration cannot be made. We know this since 35 years. For many researchers this is yet a novelty. Slowly increases the number of papers that also present toxin data. It is for me an important novelty that the Agricultural Ministry agreed in the introduction of this food and feed risk analysis into the variety registration. For this reason, these smaller or larger novelties, we hope, will help us to provide more and healthy food for the people, even they do not have an idea, how much work is behind the more and healthier food.
Comments on the Quality of English Language
Moderate editing of English language required
I checked the text and Reviewer 2 had a lot of suggestion I accepted, so I hope that the English become better. I several places I have also found mistakes to be corrected, in some cases years from authors should be omitted, etc.
Reviewer 2 Report
Comments and Suggestions for Authors
Dear author,
your review seems comprehensive with many informations inside. You've used many references that are relevant thesedays. Fusarium spp. are very serious problem for the cereals, and especially mycotoxins need extra attentions as they could be potentially very dangerous for human and animal health.
Due to that fact, I would recommend this article for publishing.
But you might think for more easier flow of reading that you re-order chapters. They all seems long and sometimes it is difficult to read as many repetitions of the text is inside. In my attachment are some typos or suggestions.
Please check the references also, there are mistakes inside.
Once more, the review is great with many new informations, that are relevant for Fusarium spp. and their roxins.

Author Response
Author's Reply to the Review Report (Reviewer 2)
Please provide a point-by-point response to the reviewer’s comments and either enter it in the box below or upload it as a Word/PDF file. Please write down "Please see the attachment." in the box if you only upload an attachment. An example can be found here.
Dear Reviewer 2.
Thank you very much for your careful work improving this review. They helped me to make better the paper and I found several other points where I realized that a clearer text would be better. Thanks also for the good opinion about this paper, based on this, I hope that it can help readers to find better solution for this very complicated syndrome we face, and a lot of research is needed to understand better. As the line numbering changed in the improved text, for each remark I gave the position of the modifications or comments to find them easily.
Thank you again, I wish you a happy Christmas,
yours sincerely
Akos Mesterhazy
* Author's Notes to Reviewer
Comments and Suggestions for Authors
Dear author,
your review seems comprehensive with many informations inside. You've used many references that are relevant thesedays. Fusarium spp. are very serious problem for the cereals, and especially mycotoxins need extra attentions as they could be potentially very dangerous for human and animal health.
Due to that fact, I would recommend this article for publishing.
But you might think for more easier flow of reading that you re-order chapters. They all seems long and sometimes it is difficult to read as many repetitions of the text is inside. In my attachment are some typos or suggestions.
Please check the references also, there are mistakes inside.
The references were controlled, the double numbering was ceased, several changes were made, the numbering corresponds to the numbers in the text. many punctuation mistakes were corrected.
I answer now the suggestions from the PDF file?
Line 25: QTL analyses was changed to QTL-analyses.
Line 40. You suggest the reorganization of the material. My problem is that by this reorganization cannot be done in this copy as the reorganization will change the citing order of the citations and names are not occurring most cases. For this an earlier version would be suitable where the names are yet included. However, the improvements in the text were made for the other reviewer, I also made improvements, and it will not be easy to follow the changes requested for the reviewers. The whole citation order should be changed, and I can do it only by hand, I do not have any means that MDPI possibly has. There is a more serious argument. When I planned the paper, I thought that after introduction the global significance of the ear rots and other losses should be presented, that the reader sees, what is on the stake. Here the mycotoxins having wide occurrence were mentioned together, causing huge losses. mostly from their occurrence in the corn production. The climate discussion is important, I positioned it to the second part of the paper. In Middle Europe the warming is clear and causes higher fumonisin and aflatoxin contamination. In North Europe and Southern Siberia, the global warming makes corn production possible that is good, but the diseases will appear also. In the Sub-Sahara Region, or the Mediterranean region large part of China or US the aflatoxin is present and will not change much, but they should be controlled by higher resistance, the mycotoxin supporting ecological conditions should be balanced by better agronomy and other practices. This chapter is therefore here, because the corn production should adapt to the local conditions, and this seems me better after the chapters and information of the resistance and other conditions were discussed. However, a higher resistance to toxigenic fungi is useful everywhere, maybe not to the same ones in each region. Considering the seasonal differences, deoxynivalenol will be an important mycotoxin, even the F. graminearum epidemics are forecasted less frequent, but 1-2 epidemics in a decade is sure to come, in some regions much more frequent. I would like to ask you to leave the present chapter ranking by this way.
Line 44. It was improved. Line 128.
Line 46. zearalenone was inserted now line 130
Line 59: explanation is given Line 143
Line 61. It is corrected. Line 146.
Line 70. Corrected, line 156
Line 97. Inserted: Bacillus thuringiensis, line 187)Line 102. I changed the literature for and aflatoxin paper. Line 190.
Line 105. Extra space delated. Line 193
- Table was modified, this seems to be OK. now. Line 223. Extra space in the bottom of the table was given.
Line 124. Maize was inserted, Corn was changed for maize in the text except two cases (sweetcorn and Corn Belt for maize, in the references I left everywhere the original version. line 231
Line 133 . It was improved to varietal Line 240.
Line 142. The sentence was rephrased. Line 250
Line 147. I inserted explanation Line 256
Line 149. I inserted grain, Line 257.
Line 152. Maize was changed. Line 274.
Line 161. Surname discarded. Line 276
Line 162. Year discarded. Line 276.
Line 171. Extra space discarded. Line 287.
Line 184. I have published in several MDPI journals, the t/ha was always good. It seems me more fit than bushel/acre. I think, the Americans have the opposite feeling. Line 301
Line 189. Extra space was inserted. Line 331.
Line 190. Maize was inserted. Line 331.
Line 198 Changed. line 341
Line 205 The sentence was rewritten, line 347-348.
Line 206 The sentence was rewritten. Line 348-350.
Line 214 It was added. Line 356
Line 218. I include a sentence about yield trial. Since 2010 we work parallel with 3-4 toxigenic species, separately, two isolates per species. The site is the Szeged nursery. For selection and breeding this is OK, for varietal release more data are needed no to make false positive or negative risk evaluation. We do not make yield tests, but we tested released top hybrids with excellent yielding ability, from the publications you can see them. Their yield data can be checked in the cited GOSZ-.VSZT tests, and before to be released they had to produce a higher yield than the checks are doing. Since 2020 we use three independent isolates. By this way, in one year we have so much data what we should work for three years with traditions one inoculum method. As the isolates act under exactly the same environmental niche, the environment isolate resistance interaction is low, when it happens, then other traits of the inoculum can influence them. We count we the means of the isolates. Then the data will be more reliable. For varietal registration three isolates and two locations are planned with three replicates, in three years for F. graminearum 54 data, and 18 means will be considered. The same data set will be present for F. verticillioides and A. flavus as well as not inoculated control with multitoxin measurement for the six DON, FUM, ZEN, AFB1, HT-2 and maybe T-2 or OTA as most important toxins.
Line 236 Improved. Line 396.
Line 255. Space deleted. Line 415.
Line 268. Full name for DON was omitted. Line 428.
Line 283-285. The sentence was rephrased. Lines 443-456.
Line 301 Good agronomy practice inserted, Line 476.
Line 312 accounted inserted, Line 478.
Line 326 Sentence improved. Line 502.
Line 332. Recorded was inserted, Line 508.
Line 342. Maize was settled. Line 519.
Line 345. Mistake improved. I checked the original data. Fusarium was not included, so I omitted. Line 521.
Line 346. Sentence accordingly modified. Line 523.
Line 367-368. Transformed to italic. Lines 549-550. .
Line 372-376. Transformed to italic. Sorry. Lines 554 and 558.
Line 385. The words “as found in” were omitted. Line 567
Line 390. The double numbers were corrected from line 572.
Line 409 comma inserted Line 594.
Line 435 commas inserted. Line 620.
Line 444. normal was changed to limit. Lines 628-629.
Line 456. the e was added to ZEN, Line 641.
Line 481. As I indicated the topic at the beginning, from didactic point of view this topic is on its place. We know something about the significance of the pathogens, what are they, what they are producing, and here based on them we can speak what the resistance means, what we know and what we should know. I intended the show the whole scenario as far as possible. I have read a number of good reviews about QTLs in resistance and so on, and I thought that this problem should be looked once from the food safety side.
Line 501. space inserted. Line 692.
Line 508 Extra space omitted. Line 699.
Line 529. grain inserted. Line 722.
Line 927. dot removed Line 1123.
Line 1219. Chapter 6.
When I see papers like Battilani or Felicia Wu, who forecast Armageddon for example of the aflatoxin problem, using a forecast system based on sophisticated climate and weather data as it would be a natural low like the 2*2=4. When I see a good resistant maize hybrid with 20% Aspergillus flavus infection and 4000 ppb aflatoxin, and besides that another one, inoculated at the same time, and infection severity far below 1%, and aflatoxin from zero to several ppb, it is clear that this difference cannot be explained. by climatic arguments. When I asked Paola, why do they not consider resistance data in forecast, she answered, we do not have such data. This was more than 10 years ago. Since that the matter did not changed MUCH. These forecasts have right when all hybrids are susceptible or very susceptible allowing a similar reaction. When the forecasts are OK in about 70-80% of the cases, this supports the general rule of more susceptible hybrids. But in 20-30% the more resistant hybrids perform better what was forecast. As nobody has data, we should create them. And then the forecast programs can include resistance level and the exactness of the forecast could go up to 90 % for example. I think, in this field we can do a lot. Many reports are about the influence of draught and heat and water economy of the soil. Under high draught and heat stress the aflatoxin in contamination can be very high. When the water supply is close to normal, even the high temperature will not prefer high aflatoxin contamination, but much less if any at all. In this respect I think, we have weapons. When we use them well, the genetic and experimental work will be able to keep or improve toxin contamination, when whether conditions otherwise would favor higher aflatoxin contamination. For this we need GAP, much better adapted hybrids and this really will help.
Line 1247. Chapter 7. I positioned this chapter here for didactic consideration, when we know, what the problem is, what are the achievements published in the literature, I thought that a methodical chapter is also necessary that explains, what the problems are we identified and how these can be managed. In a research article the methodical part is decisive, its position is different, as this is a review, the freedom can be higher. Line 1463.
Line 1274. I understand, the Rev. 1 had also problem with this figure. I changed the figure, I have now two, on A the isolates are on the x axis, on B the hybrids are on the x axis. In both subfigures the mean of the isolates or hybrids can be followed, and, in the explanation, I showed several examples, how can differ two hybrids to isolates significantly, where the means for the isolates is nearly the same. When I have only one isolate, how can I be sure that the QTLs identified are correct? Or the resistance of the given genotype does really agree with the genotype, this is the old discussion about the relation between phenotype and genotype. I think, here we have to do a lot. This is valid also for the fungicide tests, we did about 13-14 years fungicide tests for international plant protection firms, but they could not be printed as in the highly susceptible hybrids they requested to test, and these could not be protected efficiently. However, as in wheat, the fungicide testing of the much more resistant hybrids seems to be reasonable alternative that should be made yet. The more isolates were useful also here.
Line 1288. The bracket was omitted, the full stop was made. Line 1512-1513. .
Line 1294. Extra space was omitted. Line 1518.
Line 1351. I changed the text. Line 1587.
Line 1352. I inserted before yield the word grain based on your earlier remark. Line 1588.
Line 1366. The “genetically modified was inserted. Thanks Line 1606.
Line 1389. The word grain was inserted. I checked the whole text for yield, and where bit was appropriate, the grain was added. About 20-25 cases were found.
Once more, the review is great with many new informations, that are relevant for Fusarium spp. and their toxins...
Submission Date
05 December 2023
Date of this review
13 Dec 2023 10:30:31
Round 2
Reviewer 1 Report
Comments and Suggestions for Authors
Author has made a revision, which fit the journal requirement, so I recommend its acceptance.